# Fundamental bound on the persistence and capacity of short-term memory stored as graded persistent activity

Onur Ozan Koyluoglu[1]*, Yoni Pertzov[2], Sanjay Manohar[3], Masud Husain[3], Ila R Fiete[4]

[1]Department of Electrical Engineering and Computer Science, University of California, Berkeley, Berkeley, United States; [2]Department of Psychology, Hebrew University, Jerusalem, Israel; [3]Department of Experimental Psychology, University of Oxford, Oxford, United Kingdom; [4]Center for Learning and Memory, University of Texas at Austin, Austin, United States

**Abstract** It is widely believed that persistent neural activity underlies short-term memory. Yet, as we show, the degradation of information stored directly in such networks behaves differently from human short-term memory performance. We build a more general framework where memory is viewed as a problem of passing information through noisy channels whose degradation characteristics resemble those of persistent activity networks. If the brain first encoded the information appropriately before passing the information into such networks, the information can be stored substantially more faithfully. Within this framework, we derive a fundamental lower-bound on recall precision, which declines with storage duration and number of stored items. We show that human performance, though inconsistent with models involving direct (uncoded) storage in persistent activity networks, can be well-fit by the theoretical bound. This finding is consistent with the view that if the brain stores information in patterns of persistent activity, it might use codes that minimize the effects of noise, motivating the search for such codes in the brain.
DOI: https://doi.org/10.7554/eLife.22225.001

*For correspondence:
ozan.koyluoglu@berkeley.edu

Competing interests: The authors declare that no competing interests exist.

## Introduction

Short-term memory, which refers to the brain's temporary buffer of readily usable information, is considered to be a critical component of general intelligence (*Conway et al., 2003*). Despite considerable interest in understanding the neural mechanisms that limit short-term memory, the issue remains relatively unsettled. Human working memory is a complex phenomenon, involving not just short-term memory but executive selection and processing, operating on multiple timescales and across multiple brain areas (*Jonides et al., 2008*). In this study, we restrict ourselves to obtaining limits on short-term memory performance purely due to noise in persistent activity networks, if analog information is stored directly into these networks, or if it is first well-encoded to make the stored states robust to ongoing noise.

Short-term memory experiments quantify the precision of memory recall. Typically in such experiments, subjects are briefly presented with sensory inputs, which are then removed. After a delay the subjects are asked to estimate from memory some feature of the input. Consistent with everyday experience, memory *capacity* is severely limited, restricted to just a handful of items (*Miller, 1956*), and recall performance is worse when there are more items to be remembered. *Persistence* can also be limited, though forgetting over time is a less severe constraint than capacity: several experiments show that recall performance declines with delay (*Luck and Vogel, 1997*; *Jonides et al., 2008*; *Barrouillet et al., 2009*; *Barrouillet et al., 2011*; *Barrouillet et al., 2012*; *Pertzov et al.,*

*2013*; *Wilken and Ma, 2004*; *Bays et al., 2011*; *Pertzov et al., 2017*; *Anderson et al., 2011*), at least when many items are stored in memory.

Efforts in experimental and theoretical psychology to understand the nature of these memory constraints (*Atkinson and Shiffrin, 1968*) have led to quantification of human memory performance, and to phenomenological models that can fit limitations in capacity (*Zhang and Luck, 2008*; *Bays and Husain, 2008*; *van den Berg et al., 2012*) or in persistence (*Wilken and Ma, 2004*; *Barrouillet et al., 2012*). They have also led to controversy: about whether memory consists of discrete 'slots' for a limited maximum number of items (*Miller, 1956*; *Cowan, 2001*; *Zhang and Luck, 2008*) or is more continuously allocable across a larger, variable number of items (*van den Berg et al., 2012*; *Bays and Husain, 2008*); about whether forgetting in short-term memory can be attributed in part to some inherent temporal decay of an activity or memory variable over time (*Barrouillet et al., 2012*; *Campoy, 2012*; *Ricker and Cowan, 2014*; *Zhang and Luck, 2009*) or is, as more widely supported, primarily due to interference across stored items (*Lewandowsky et al., 2009*).

These controversies have been difficult to resolve in part because different experimental paradigms lend support to different models, while in some cases the resolution of memory performance data is not high enough to adjuciate between models. In addition, psychological models of memory performance make little contact with its neural underpinnings; thus, it is difficult to mediate between them on the basis of mechanism or electrophysiological studies.

On the mechanistic side, persistent neural activity has been widely hypothesized to form the substrate for short-term memory. The hypothesis is based on a corpus of electrophysiological work establishing a link between short-term memory and persistent neural activity (*Funahashi, 2006*; *Smith and Jonides, 1998*; *Wimmer et al., 2014*). Neural network models of analog persistent activity predict a degradation of information over time (*Compte et al., 2000*; *Brody et al., 2003*; *Boucheny et al., 2005*; *Burak and Fiete, 2009*; *Fung et al., 2010*; *Mongillo et al., 2008*; *Burak and Fiete, 2012*; *Wei et al., 2012*), because of noise in synaptic and neural activation. If individual analog features are assumed to be directly stored as variables in such persistent activity networks, the time course of degradation of persistent activity should directly predict the time course of degradation in short-term memory performance. However, these models do not typically consider the direct storage of multiple variables (but see (*Wei et al., 2012*) ), and in general their predictions have not been directly compared against human psychophysics experiments in which the memory load and delay period are varied.

In the present work, we make the following contributions: (1) Generate psychophysics predictions for information degradation as a function of delay period and number of stored items, if information is stored directly, without recoding, in persistent activity neural networks of a fixed total size; (2) Generate psychophysics predictions (though the use of joint source-channel coding theory) for a model that assumes information is restructured by encoding and decoding stages before and after storage in persistent activity neural networks; (3) Compare these models to new analog measurements (*Pertzov et al., 2017*) of human memory performance on an analog task as the demands on both maintenance duration and capacity are varied.

We show that the direct storage predictions are at odds with human memory performance. We propose that noisy storage systems, such as persistent activity networks, may be viewed as noisy channels through which information is passed, to be accessed at another time. We use the theory of *channel coding* and *joint source-channel coding* to derive the information-theoretic upper-bound on the achievable accuracy of short-term memory as a function of time and number of items to be remembered, assuming a core of graded persistent activity networks. According to the channel coding view, the brain might strategically restructure information before storing it, to use the available neurons in a way that minimizes the impact of noise upon the ability to retrieve that information later. We apply our framework, which requires the assumption of additional encoding and decoding stages in the memory process, to psychophysical data obtained using the technique of delayed estimation (*Ma et al., 2014*), which provides a sensitive measure of short-term memory recall using a continuous, analog response space, rather than discrete (Yes/No) binary recall responses.

We show that empirical results are in substantially better agreement with the functional form of the theoretical bound than with predictions from a model of direct storage of information in persistent activity networks.

Our treatment of the memory problem is distinct from other recent approaches rooted in information theory (*Brady et al., 2009*; *Sims et al., 2012*), which consider only *source coding* – they assume that internal representations have a limited number of states, then compute the minimal distortion achievable in representing an analog variable with these limited states, after redundancy reduction and other compression. All representations are noise-free. By contrast, our central focus is precisely on noise and its effects on memory degradation *over time*, because the stored states are assumed to diffuse or random-walk across the set of possible stored states. The emphasis on representation with noise involves *channel coding* as the central element of our analysis.

Our present work is also complementary to efforts to understand short-term memory as rooted in variables other than persistent activity, for instance the possibility that short-term synaptic plasticity, through facilitation (*Mongillo et al., 2008*; *Barak and Tsodyks, 2014*; *Mi et al., 2017*), might 'silently' (*Stokes, 2015*) store short-term memory, which is reactivated and accessed through intermittent neural activity (*Lundqvist et al., 2016*).

## Results

### Analog measurement of human short-term memory

We consider data from subjects performing a delayed estimation task (*Figure 1—source data 1*). We briefly summarize the paradigm and the main findings; a more detailed description can be found in *Pertzov et al. (2017)* Subjects view a display with several ($K$) differently colored and oriented bars that are subsequently removed for the storage (delay) period. Following the storage period, subjects were cued by one of the colored bars in the display, now randomly oriented, and asked to rotate it to its remembered orientation. Bar orientations in the display were drawn randomly from the uniform distribution over all angles (thus the range of orientations lies in the circular interval $[0, \pi]$) and the report of the subject was recorded as an analog value, to allow for more detailed and quantitative comparisons with theory (*van den Berg et al., 2012*). Importantly, both the number of items ($K$) and the storage duration ($T$) were varied.

When only a single item had to be remembered, the length of the storage interval had no statistically significant influence on the distribution of responses over the intervals considered (*Figure 1B*, with different delays marked by different shades and line styles; errors <10 degrees, effect of delay: $F(3, 36) = 1.3, p = 0.3$; errors between $30 - 50$ degrees: $F(3, 36) = 0.2, p = 0.9$). By contrast, response accuracy degraded significantly with delay duration when there were 6 items in the stimulus (*Figure 1C*; true orientation subtracted from all responses to provide a common center at 0 degrees). The number of very precise responses decreased (errors <10 degrees, effect of delay: $F(3, 36) = 6.15, p = 0.002$), with a corresponding increase in the number of trials with large errors (e.g. errors between $30 - 50$ degrees, effect of delay: $F(3, 36) = 5.4, p = 0.004$).

Overall, the squared error in recalling an item's orientation (*Figure 1D*), averaged over subjects, increased with delay duration ($F(3, 27) = 49, p<0.001$) and also with item number ($F(3, 27) = 48, p<0.001$). The data show a clear interaction between storage interval duration and set size ($F(9, 81) = 17, p<0.001$), apparent as steeper degradation slopes for larger set-sizes. In summary, for a small number of items (e.g. $K = 1, 2$), increasing the storage duration does not strongly affect performance, but for any fixed delay, increasing item number has a more profound effect.

Finally, at all tested delays and item numbers, the squared errors are much smaller than the squared range of the circular variable, and any sub-linearities in the curves cannot be attributed to the inevitable saturation of a growing variance on a circular domain (*Figure 1—figure supplement 1*).

### Information degradation in persistent activity networks

In this and all following sections, we start from the hypothesis that persistent neural activity underlies short-term information storage in the brain. The hypothesis is founded on evidence of a relationship between the stored variable and specific patterns of elevated (or depressed) neural activity (*Taube, 1998*; *Aksay et al., 2001*) that persist into the memory storage period and terminate when the task concludes, and on findings that fluctuations in delay-period neural activity can be predictive of variations in memory performance (*Funahashi, 2006*; *Smith and Jonides, 1998*; *Blair and Sharp,*

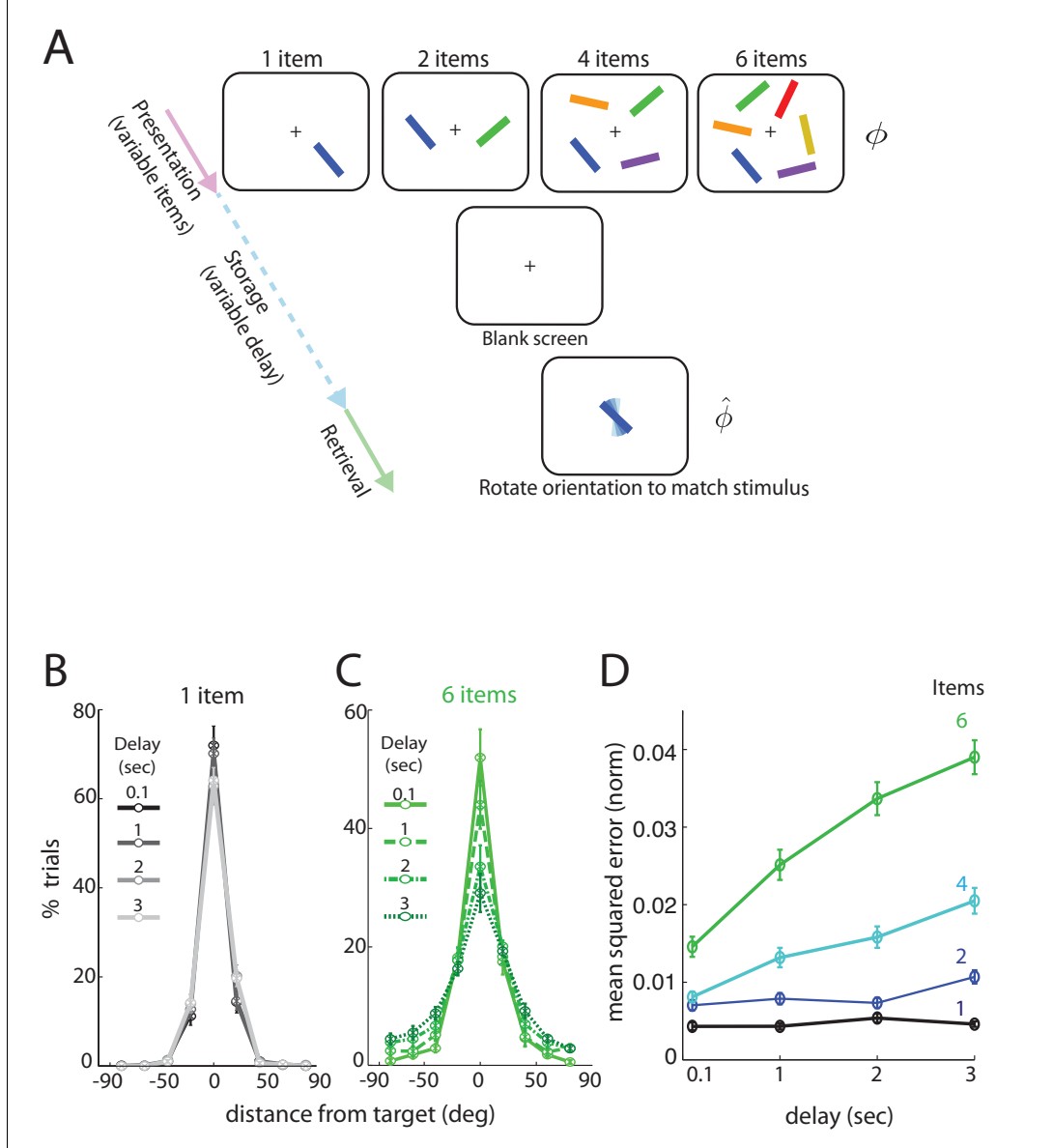

**Figure 1.** Human performance on an analog delayed orientation matching task with variable item number and storage duration. (**A**) Setup of a delayed orientation estimation task to probe human short-term memory. A variable number of bars with different colors and uniformly randomly drawn orientations are presented for 500 msec. Following a variable delay, the subjects are asked to adjust the orientation of a cue bar, by using a dial, to match the remembered orientation of the bar of the same color from the presentation. (**B**) Distribution of responses for one item, plotted so the target orientation is centered at zero. Different shades and line styles represent different delays. Note that responses did not vary significantly with storage duration. (**C**) Distribution of responses for six items varies with storage duration. (**D**) Mean squared error of recall on the task of *Figure 1A* (averaged across subjects and trials, and normalized by $(180°)^2$, the square of the range of the stored variable), as item number and delay duration are systematically varied. Error bars denote SEM across participants.

DOI: https://doi.org/10.7554/eLife.22225.002

The following source data and figure supplement are available for figure 1:

**Source data 1.** Experiment data used in the manuscript.
DOI: https://doi.org/10.7554/eLife.22225.004

**Figure supplement 1.** Similar variance statistics for bounded versus unbounded domains over range relevant for performance data.
DOI: https://doi.org/10.7554/eLife.22225.003

*1995*; *Miller et al., 1996*; *Romo et al., 1999*; *Supèr et al., 2001*; *Harrison and Tong, 2009*; *Wimmer et al., 2014*).

Neural network models like the ring attractor generate an activity bump that is a steady state of the network and thus persists when the input is removed, *Figure 2A*. All rotations of the canonical activity bump form a one-dimensional continuum of steady states, *Figure 2B*. Relatively straightforward extensions of the ring network can generate 2D or higher-dimensional manifolds of persistent states. However, any noise in network activity, for instance in form of stochastic spiking (*Softky and Koch, 1993*; *Shadlen and Newsome, 1994*), leads to lateral random drift along the manifold in the form of a diffusive (Ornstein-Uhlenbeck) random walk (*Compte et al., 2000*; *Brody et al., 2003*;

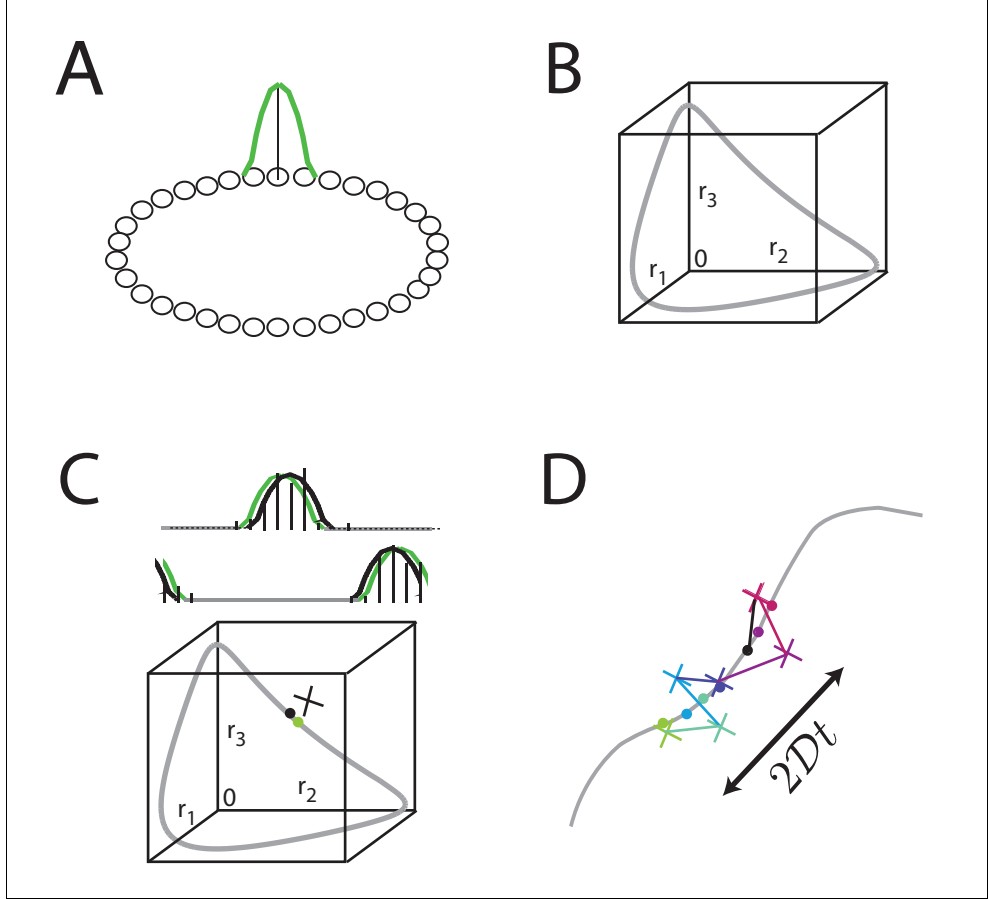

**Figure 2.** Analog persistent activity networks and information decay over time. (**A**) In a ring network, each neuron excites its immediate neighbors and inhibits all the rest (weight profiles not shown). A single bump of activity (green) is a steady state of such a network of such a network, as are all its translations around the ring. (**B**) A 'state-space' view of activity in the ring network: each axis represents the activity of one neuron in the network; if there are $N$ neurons in the network, this state-space plot is $N$-dimensional. Any point inside the state space represents some possible instantaneous configuration of activity in the $N$ neurons. The grey curve represents the set of steady states, which traces a 1-dimensional manifold because the stable states are just translations of a canonical activity bump along a single dimension. (**C**) Top: Grey: a schematic non-noisy activity bump; black vertical lines: schematic spikes emitted by neurons after the state is initialized according to the grey curve. Black curve: A best-fit activity profile for the emitted spikes is shifted relative to the original grey bump simply because of the stochastic spikes. Bottom: the state space view of (**B**), with the addition of the state corresponding to the non-noisy initial activity bump (grey filled circle), the noisy spiking state (black cross), and the projection of the noisy spiking state to the best-fit or closest non-noisy activity profile (black filled circle). (**D**) Over longer periods of time, activity fluctuations seen in (**C**) drive a diffusive drift (random walk) along the manifold of stable states, with a squared error that grows linearly with time.
DOI: https://doi.org/10.7554/eLife.22225.005

*Boucheny et al., 2005*; *Wu et al., 2008*; *Burak and Fiete, 2009*; *Fung et al., 2010*; *Burak and Fiete, 2012*), *Figure 2C–D*.

A defining feature of such random walks is that the squared deviation of the stored state relative to its initial value will grow linearly with elapsed time over short times, *Figure 2D*, with a proportionality constant $2\mathcal{D}$ (where $\mathcal{D}$ is the diffusivity) that depends on quantities like the size of the network and the peak firing rate of neurons (*Burak and Fiete, 2012*).

## Memory modeled as direct storage in persistent activity networks

Suppose that the variables in a short-term memory task were *directly* transferred to persistent activity neural networks with a manifold of fixed points that matched the topology of the represented variable. Thus, $K$ circular variables would be stored, entry-by-entry, in $K$ 1-dimensional (1D) ring networks (*Ben-Yishai et al., 1995*). (Alternatively, the $K$ variables could be stored in a single network with a $K$-dimensional manifold of stable states, as described in the Appendix; the performance in neural costs and in fit to the data of this version of direct storage is worse than with storage in $K$ 1D networks, thus we focus on banks of 1D networks.)

When $N$ neural resources (e.g. composed of $N$ sets of $M$ neurons each, for a total of $NM$ neurons) are split into $K$ networks, each network is left with $N/K$ resources ($NM/K$ neurons in our example) for storage of a 1D variable. We know from (*Burak and Fiete, 2012*) that the diffusivity of the state in each of these 1D persistent activity networks will scale as the inverse of the number of neurons and of the peak firing rate per neuron. In other words, the diffusion coefficient is given by $\bar{\mathcal{D}}(K,N) = \mathcal{D}K/N$, where $\mathcal{D}$ is a diffusivity parameter independent of $K, N$ (but $\mathcal{D} \propto 1/M$). So long as the squared error remains small compared to the squared range of the variable, it will grow linearly in time at a rate given by $2\bar{\mathcal{D}}(K,N)$ (indeed, in the psychophysical data, the squared error remains small compared to the squared range of the angular variable; see *Figure 1—figure supplement 1*). Therefore the mean squared error (MSE) is given by:

$$D_{\mathrm{MSE}}(\Phi, K, T) = \Phi^2 \frac{2\mathcal{D}K}{N} T. \tag{1}$$

The only free parameter in the expression for MSE as a function of time and item number is the ratio $N/2\mathcal{D}$. Because the inverse diffusivity parameter $1/\mathcal{D}$ scales with the number of neurons ($M$ in our example) when $N, K$ are held fixed, the product $N/(2\mathcal{D})$ is proportional to the total number of neurons ($N/(2\mathcal{D}) \propto NM$). This ratio therefore functions as a combined neural resource parameter.

## Direct storage is a poor model of memory performance

To fit the theory of direct storage to psychophysics data, we find a single best-fit value (with weighted least-squares) of the free parameter $N/2\mathcal{D}$ across all item numbers and storage durations. For each item number curve, the fits are additionally anchored to the shortest storage period point ($T = 100$ ms), which serves as a proxy for *baseline* performance at zero delay. Such baseline errors close to zero delay – which may be due to limitations in sensory perception, attentional constraints, constraints on the rate of information encoding (loading) into memory, or other factors – are not the subject of the present study, which seeks to describe how performance will *deteriorate over time* relative to the zero-delay baseline, as a function of storage duration and item number.

As can be seen in *Figure 3A*, the direct storage theory provides a poor match to human memory performance ($p$ values that the data occur by sampling from the model, excluding the 100 ms timepoint: $0.07, 0.38, <10^{-4}$ for 1 item; $0.39, <10^{-4}, 0.2$ for 2 items; $0.09, 0.29, 0.08$ for 4 items, and $<10^{-3}, <10^{-4}, <10^{-4}$ for 6). These $p$-values strongly suggest rejection of the model.

Does the direct storage model fail mostly because its dependence on time and item number are linear, while the data exhibits some nonlinear effects at the largest delays? On the contrary, direct storage fails to fit the data even at short delays when the performance curves are essentially linear (see the systematic underestimation of squared error by the model over $\leq 2$ second delays in the 4- and 6-item curves). If anything, the slight sub-linearity in the 6-item curve at longer delays tends to bring it closer to the other curves and thus to the model, thus its effect is to slightly reduce the discrepancy between the data and fits from direct storage theory.

One view of the results, obtained by selecting model parameters to best match the 6-item curve, is that direct storage theory predicts an insufficiently strong *improvement* in performance with

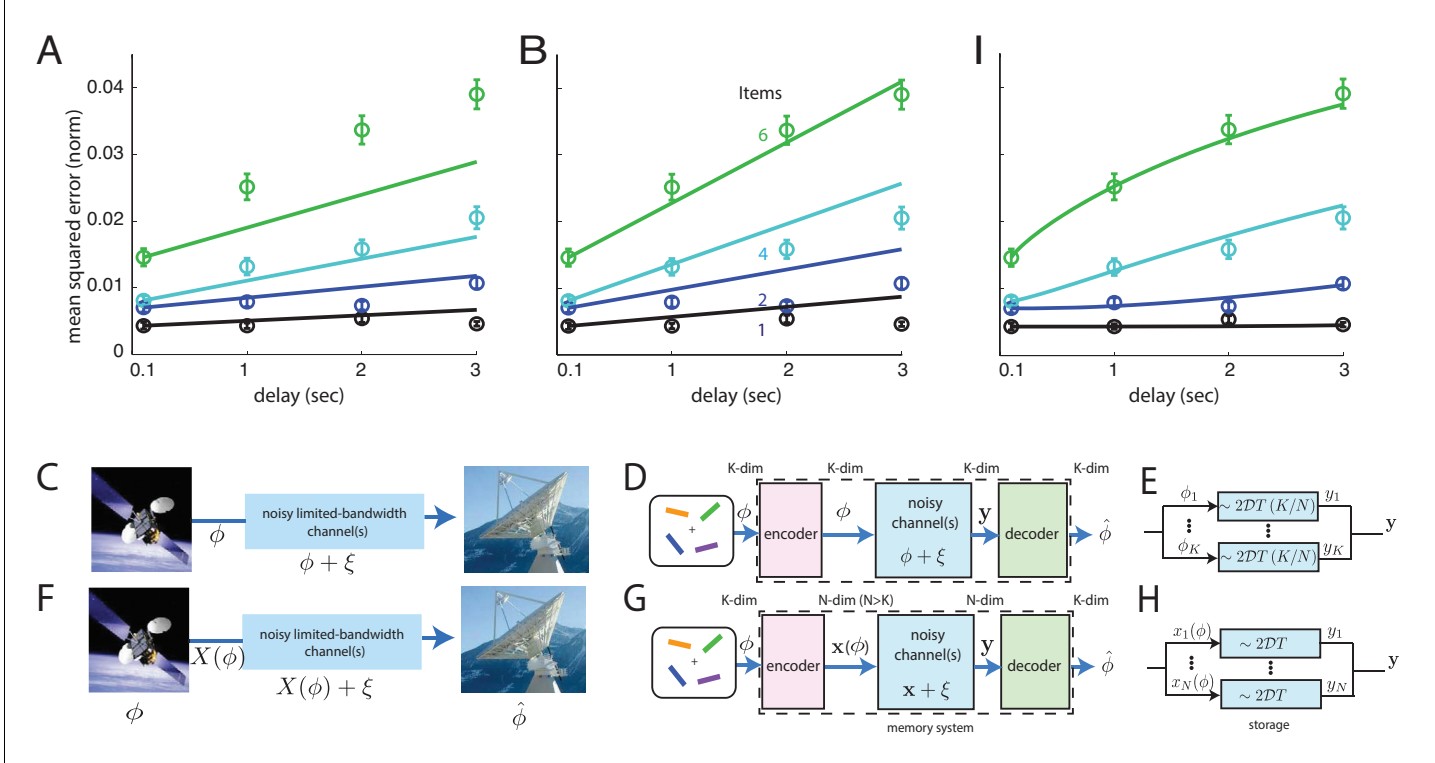

**Figure 3.** Comparison of direct and coded storage models using persistent activity networks with human memory performance. (A) Lines: predictions from the direct storage model for human memory. The theory specifies all curves with a single free parameter, after shifting each curve to the measured value of performance at the shortest delay interval of 100 ms. Fits performed by weighted least squares (weights are inverse SEM). (B) Similar to (A), but parameters fit by ordinary least-squares to only the 6-item curve; note the discrepancy in the 1- and 2-item fits. (C–E) Information ($\phi$) is directly transmitted (or stored) in a noisy channel, and at the end an estimate of $\hat{\phi}$ of $\phi$ is recovered. (C) A scenario involving space-to-earth communication. (D) The scenario for direct storage in noisy memory banks (the nosy channels); the encoder and decoder are simply the identity transformation in the case of direct storage and hence do nothing. (E) The $K$ pieces of information in the $K$-dimensional vector $\phi$ are each represented in one of $K$ continuous attractor neural networks of size $N/K$ neurons each. Each attractor representation accumulates squared error linearly over time and inversely with $N/K$. (F–H) Same as (C–E), but here information is first encoded ($\phi \rightarrow \mathbf{X}(\phi)$) with appropriate structure and redundancy to combat the channel noise. A good encoder-decoder pair can return an estimate $\hat{\phi}$ that has lower error than the direct strategy, even with similar resource use, mitigating the effects of channel noise for high-fidelity information preservation. (H) The $K$-dimensional $\phi$ is encoded as the ($N$-dimensional) codeword $\mathbf{x}$, each entry of which is stored in one of $N$ persistent activity networks. Squared error in the channel grows linearly with time as before; however, the resources used to build $K$ channels of quality $(N/K)1/2\mathcal{D}$ from before are redirected into building $N$ channels of poorer quality $1/2\mathcal{D}$ (assuming $N>K$). The decoder estimates $\phi$ from $N$-dimensional output $\mathbf{y}$. (I) Same as (A), but the model lines are the lower-bound on mean-squared error obtained from an information-theoretic model of memory with good coding. (Model fit by weighted least-squares; the theory specifies all curves with two free parameters, after shifting each curve to the measured value of performance at the shortest delay interval of 100 ms).

DOI: https://doi.org/10.7554/eLife.22225.006

The following figure supplements are available for figure 3:

**Figure supplement 1.** Cross-validated comparison of the direct and well-coded storage models after leaving out $T = 1$s datapoints.

DOI: https://doi.org/10.7554/eLife.22225.007

**Figure supplement 2.** Cross-validated comparison of the direct and well-coded storage models after leaving out $T = 2$s datapoints.

DOI: https://doi.org/10.7554/eLife.22225.008

**Figure supplement 3.** Comparison of models after removal of the shortest (100 ms) delay time-point under the argument that it represents a different memory process (iconic memory).

DOI: https://doi.org/10.7554/eLife.22225.009

**Figure supplement 4.** Redefining item numbers as $K = [1 >4 >8 >12]$ (instead of $K = [1 >2 >4 >6]$) to take into account the memorization of item color in addition to orientation.

DOI: https://doi.org/10.7554/eLife.22225.010

decreasing item number, *Figure 3B* (*p*-values for direct-storage model when fit to the 6-item responses: $<10^{-3}, 10^{-3}, <10^{-4}$ for 1 item; $<10^{-2}, <10^{-4}, <10^{-4}$ for 2 items; $0.76, <10^{-2}, 2 \times 10^{-3}$ for 4 items; $0.22, 0.39, 0.38$ for 6, excluding the 100 ms delay time-point; the *p*-values for the 1- and 2-item curves strongly suggest rejection of the model).

## Information-theoretic bound on memory performance with well-coded storage

Even if information storage in persistent activity networks is a central component of short-term memory, describing the storage step is not a sufficient account of memory. This fact is widely appreciated in memory psychophysics, where it has been observed that variations in attention, motivation, and other factors also affect memory performance (*Atkinson and Shiffrin, 1968*; *Matsukura et al., 2007*). Here we propose that, even discounting these complex factors, direct storage of a set of continuous variables into persistent activity networks with the same total dimension of stable states lacks generality as a model of memory because it does not consider how pre-encoding of information could affect its subsequent degradation, *Figure 3C–E*. This omission could help account for the mismatch between predictions from direct storage and human behavior, *Figure 3A–B*.

Storing information in noisy persistent activity networks means that after a delay there will be some information loss, as described above. Mathematically, information storage in a noisy medium is equivalent to passing the information through a noisy information channel. To allow for high-fidelity communication through a noisy channel, it is necessary to first appropriately encode the signal, *Figure 3F*. Encoding for error control involves the addition of appropriate forms of redundancy tailored to the channel noise. As shown by Shannon (*Shannon, 1948*), very different levels of accuracy can be achieved with different forms of encoding for the same amount of coding redundancy and channel noise. Thus, predictions for memory performance after good encoding may differ substantially from the predictions from direct storage even though the underlying storage networks (channels) are identical.

Thus, a more general theory of information storage for short-term memory in the brain would consider the effects of arbitrary encoder-decoder pairs that sandwich the noisy storage stage, *Figure 3G*. In such a three-stage model, information to be stored is first passed to an encoder, which performs all necessary encoding. Encoding strategies may include source coding or compression of the data as well as, critically, channel coding — the addition of redundancy tailored to the noise in the channel so that, subject to constraints on how much redundancy can be added, the downstream effects of channel noise are minimized (*Shannon, 1948*). The coded information is stored in persistent activity networks, *Figure 3H*. Finally, the information is accessed by a decoder or readout, *Figure 3G*. Here, we derive a bound on the best performance that can be achieved by any coding or decoding strategy, if the storage step involves graded persistent activity.

The encoder transforms the $K$-dimensional input variable into an $N$ dimensional codeword, to be stored in a bank of storage networks with an $N$-dimensional manifold of persistent activity states (in the form of $N$ networks with a 1-dimensional manifold each, or 1 network with an $N$-dimensional manifold, or something in between). To equalize resource use for the persistent activity networks in both direct storage and coded storage models of memory, the $N$ stored states have a diffusivity $\mathcal{D}$ each, in contrast to the diffusivity of $\mathcal{D}K/N$ each for $K$ states (compare *Figure 3D–E and and G–H*). The storage step is equivalent to passage of information through additive Gaussian information channels, with variance proportional to the storage duration $T$ and to the diffusivity. The decoder error-corrects the output of the storage stage and inverts the code to provide an estimate of the stored variable. (For more details, see Materials and methods and Appendix.)

We can use information theory to derive the *minimum achievable* recall error over all possible encoder-decoder structures, for the given statistics of the variable to be remembered and the noise in the storage information channels. In particular, we use *joint source-channel coding* theory to first consider at what rate information can be conveyed through a noisy channel for a given level of noise and coding redundancy, then obtain the minimal achievable distortion (recall error) for that information rate (see Materials and Appendix). We obtain the following lower-bound on the recall error:

$$D_{\mathrm{MSE}}(\Phi, K, T) = \frac{\Phi^2}{2\pi e}\left(1 + \frac{1}{2\mathcal{D}T}\right)^{-N/K} \qquad (2)$$

This result is the *theoretical lower bound* on MSE achievable by any system that passes information through a noisy channel with the specified statistics: a Gaussian additive channel noise of zero mean and variance $2\mathcal{D}T$ per channel use, a codeword of dimension $N$, and a variable to be transmitted (stored) of dimension $K$, with entries that lie in the range $[0, \Phi]$. The bound becomes tight asymptotically (for large $N$), but for small $N$ it remains a strict lower-bound. Although the potential for decoding errors is reduced at smaller $N$, the qualitative dependence of performance on item number and delay should remain the same (Appendix and (*Polyanskiy et al., 2010*) ). The bound is derived by dividing the total resources (defined here, as in the direct storage case, as the ratio $N/2\mathcal{D}$) evenly across all stored items (details in Appendix), similar to a 'continuous resource' conception of memory. The same theoretical treatment will admit different resource allocations, for instance, one could split the resources into a fixed number of pieces and allocate those to a (sub)set of the presented items, more similar to the 'discrete slots' model.

A heuristic derivation of the result above can be obtained by first noting that the capacity of a Gaussian channel with a given signal-to-noise ratio ($SNR$) is $I_{Gauss} = \frac{1}{2}\log(1 + SNR)$. The summed capacity of $N$ channels, spread across the $K$ items of the stored variable, produces $I_{per\ item} = \frac{N}{K}I_{Gauss}$. The variance of a scalar within the unit interval represented by $I$ bits of information is bounded below by $e^{-2I}$. Inserting $I_{per\ item}$ into the variance expression and $SNR = 1/2\mathcal{D}T$ into $I_{Gauss}$, yields **Equation 2** , up to scaling prefactors. The Appendix provides more rigorous arguments that the bound we derive is indeed the best that can theoretically be achieved.

*Equation 2* exhibits some characteristic features, including, first, a joint dependence on the number of stored items and the storage duration. According to this expression, the time-course of memory decay depends on the number of items. This effect arises because items compete for the same limited memory resources and when an item is allocated fewer resources it is more susceptible to the effects of noise over time. Second, the scaling with item number is qualitatively different than the scaling with storage duration: Increasing the number of stored items degrades performance much more steeply than increasing the storage interval, because item number is in the exponent. For a single memorized feature or item, the decline in accuracy with storage interval duration is predicted to be weak. On the other hand, increasing the number of memorized items while keeping the storage duration fixed should lead to a rapid deterioration in memory accuracy.

We next consider whether the performance of an optimal encoder (given this lower bound) can be distinguished from the direct storage model based on human performance data. The two predictions differ in their dependence upon the number of independent storage channels or networks, $N$, which we do not know how to control in human behavior. Equally important, since *Equation 2* provides a theoretical limit on performance, it is of interest to learn whether human behavior approximates the limit, and where it might deviate from it.

## Comparison of theoretical bound with human performance

In comparing the psychophysical data to the theoretical bound on short-term memory performance, there are two unknown parameters, $1/2\mathcal{D}$ (the inverse diffusivity in each persistent activity network) and $N$ (the number of such networks), both of which scale linearly with the neural resource of neuron number. The product of these parameters corresponds to total neural resource exactly as in the direct storage case. We fit *Equation 2* to human performance data, assuming as in the direct storage model that the total neural resource is fixed across all item numbers and delay durations, and setting the 100 ms delay values of the theoretical curves to their empirical values.

The resulting best fit between theory and human behavior is excellent (*Figure 4E*; $p$ values that the data means may occur by sampling from the model, excluding the $T = 100$ ms time-points: $0.99, 0.07, 0.75$ for 1 item; $0.46, 0.07, 0.60$ for 2 items; $0.54, 0.24, 0.43$ for 4 items; $0.89, 0.38, 0.32$ for 6; all values are larger than 0.05, most much more so. These $p$ values indicate a significantly better fit to data than obtained with the direct storage model).

If we penalize the well-coded storage model for its extra parameter compared to direct storage ($1/2\mathcal{D}$ and $N$, versus the single parameter $\mathcal{D}/N$ for the direct storage model) through the Bayesian Information Criterion (BIC), a likelihood-based hypothesis comparison test (that more stringently penalizes model parameters than the AIC or Aikike Information Criterion), the evidence remains very strongly in favor of the well-coded memory storage model compared to direct storage ($\Delta\mathrm{BIC} \approx 99 \gg 10$, where 10 is the cutoff for 'very strong' support) (*Kass and Raftery, 1995*). In fact,

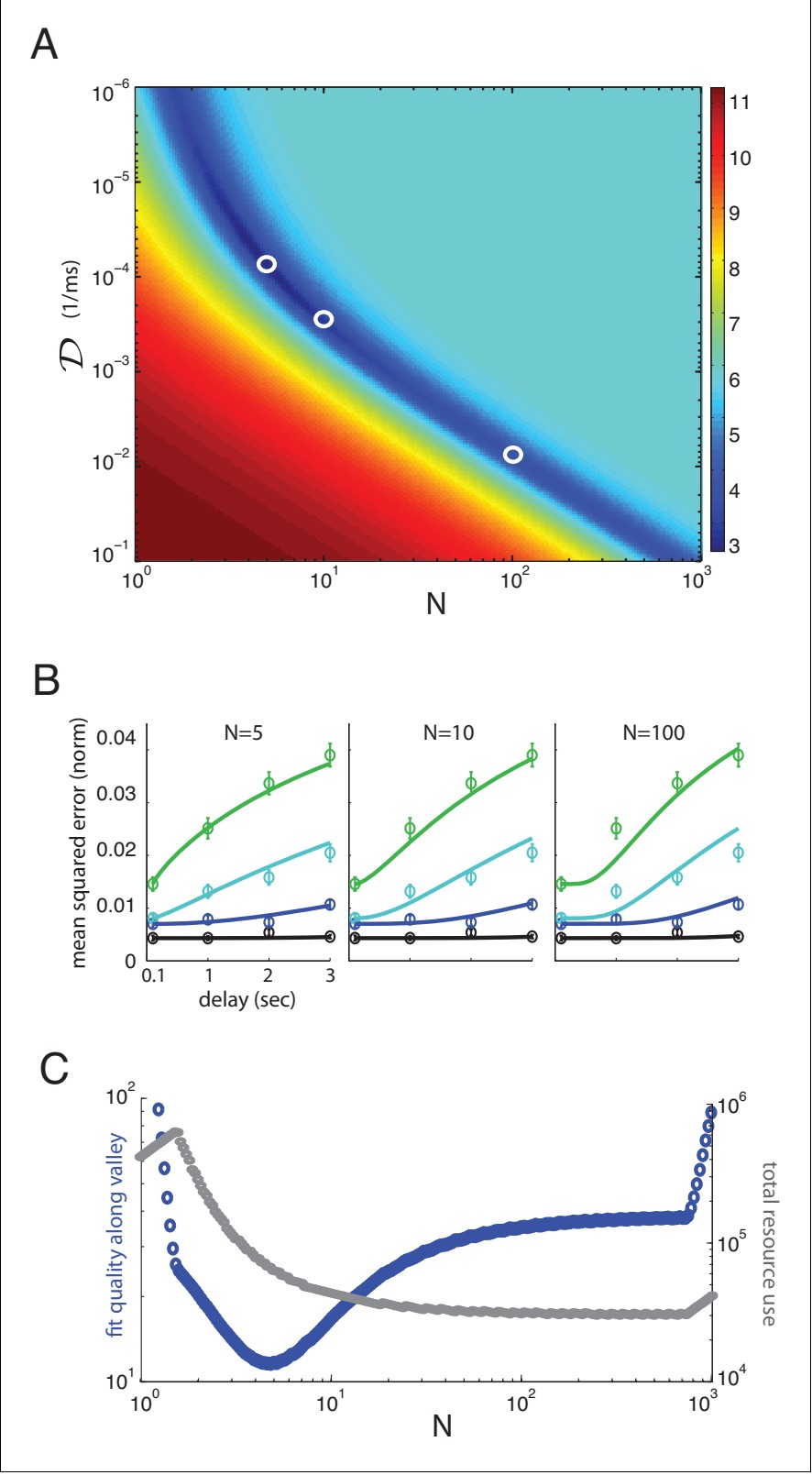

**Figure 4.** Multiplicity of reasonable parametric solutions for the well-coded storage model, with $N = 5 - -10$ networks providing the best fits to human performance. (**A**) The weighted least-squares error (colorbar indicating size of error on right) of the well-coded model fit to psychophysics data as a function of the two fit parameters, $\mathcal{D}$ and $N$. The deep blue valley running near the diagonal of the parameter space constitutes a set of reasonable fits

*Figure 4 continued on next page*

*Figure 4 continued*
to the data. (B) Three fits to the data using parameters along the valley, sampled at $N = 5, 10$, and $100$. These three parameter sets are indicated by white circles in (A). (C) Blue curve: the weighted least-squares error in the fit between data and theory along the bottom of the valley seen in (A). Gray curve: the total resource use for the corresponding points along the valley.
DOI: https://doi.org/10.7554/eLife.22225.011
The following figure supplements are available for figure 4:

**Figure supplement 1.** Performance of individual subjects and fits to well-coded storage model.
DOI: https://doi.org/10.7554/eLife.22225.012
**Figure supplement 2.** Fits of individual subject performance to direct storage model with hypothesis comparison score between direct and well-coded storage models.
DOI: https://doi.org/10.7554/eLife.22225.013

according to the BIC, the discrepancy in the quality of fit to the data between the models is so great that the increased parameter cost of the well-coded memory model barely perturbs the evidence in its favor. Some more statistical controls by jackknife cross-validation of the two models (*Figure 3—figure supplement 1*, *Figure 3—figure supplement 2*), exclusion of the $T = 100$ ms point on the grounds that it might represent iconic memory recall rather than short-term memory (*Figure 3—figure supplement 3*), and redefinition of the number of items in memory to take into account the colors and orientations of the objects are given in the Appendix (*Figure 3—figure supplement 4*); the results are qualitatively unchanged, and also do not result in large quantitative deviations in the extracted parameters (discussed below).

The two-dimensional parameter space for fitting the theory to the data contains a one-dimensional manifold of reasonable solutions, *Figure 4A* (dark blue valley), most of which provide better fits to the data than the direct storage model. Some of these different fits to the data are shown in *Figure 4B*. At large values of $N$, the manifold is roughly a hyperbola in $\log N$ and $\log(1/2\mathcal{D})$, suggesting that the logarithms of the two neural resource parameters can roughly trade off with each other; indeed, the total resource use in the one-dimensional solution valley is roughly constant at large $N$, *Figure 4C* (gray curves). However, at smaller $N$, the resource use drops with increasing $N$. The fits are not equally good along the valley of reasonable solutions, and the best fit lies near $N = 5$ independent networks or channels (for jackknife cross-validation fits, see *Figure 3—figure supplement 1*, *Figure 3—figure supplement 2*, the best fits for the coded model can be closer to $N = 10$; thus, the figure obtained for the number of memory networks should be taken as an order-of-magnitude estimate rather than an exact value). Resource use in the valley declines with increasing $N$ to its asymptotic constant value (thus larger $N$ would yield bigger representational efficiencies); however, by $N = 5$, resource use is already close to its final asymptotic value, thus the gains of increasing the number of separate memory networks beyond $N = 5 - 10$ diminish. The theory also provides good fits to individual subject performance for all ten subjects, using parameter values within a factor of 10 (and usually much less than a factor of 10) of each other (see Appendix).

## Comparison of neural resource use in direct and well-coded storage models of memory

Finally, we compare the neural resources required for storage in the direct storage model (best-fit) compared to the well-coded storage model. We quantify the neural resources required for well-coded storage as the product of the number of networks $N$ with the inverse diffusive coefficient $1/2\mathcal{D}$. This is proportional to the number of neurons required to implement storage. To replicate human behavior, coded storage requires resources totaling $N/2\mathcal{D} \approx 32$ (in units of seconds) for $N = 5$, and $N/2\mathcal{D} \approx 22$ (s) for $N = 10$, corresponding to the parameter settings for the fits in *Figures 4C* and 5B (center), respectively. By contrast, uncoded storage requires a 40-fold increase in $N$ or a 40-fold decrease in the diffusive growth rate in squared error, $2\mathcal{D}$, per network (or a corresponding increase in the product, $N/2\mathcal{D}$), because $N/2\mathcal{D} \approx 1215$ (s) under direct storage, to produce the best-fit result of *Figure 3A*. Thus, well-coded storage requires substantially fewer resources in the persistent activity networks for similar performance (assuming best fits of each produce similar performance). Equivalently, a memory system with good encoding can achieve substantially better performance with the same total storage resources, than if information were directly stored in persistent activity networks.

This result on the disparity in resource use between uncoded and coded information storage is an illustration of the power of strong error-correcting codes. Confronted with the prospect of imperfect information channels, finitely many resources, and the need to store or transmit information faithfully, one may take two different paths.

The first option is to split the total resources into $K$ storage bins, into which the $K$ variables are stored; when there are more variables, there are more bins and each variable receives a smaller bin. The other is to store $N$ quantities in $N$ bins regardless of $K$, by splitting each of the $K$ variables into $N$ pieces and assigning a piece from each of the different variables to one bin; when there are more variables, each variable gets a smaller piece of the bin. In the former approach, which is similar to the direct storage scenario, increasing $N$ would lead to improvements in the fidelity of each of the $K$ channels, *Figure 4D*. In the latter approach, which is the strong coding strategy, increasing $N$ would increase the number of channels while keeping their fidelity fixed, *Figure 4B*. The latter ultimately yields a more efficient use of the same total resources in terms of the final quality of performance, especially for larger values of $N$, at least without considering the cost of the encoding and decoding steps.

If we hold the total resource $N/2\mathcal{D} \propto NM$ fixed, the lowest achievable MSE (*Equation 2*) in the well-coded memory model is reached for maximally large $N$ and thus maximally large $\mathcal{D}$. However, human memory performance appears to be best-fit by $N \sim 10$. It is not clear, if our model does capture the basic architecture of the human memory system, why the memory system might operate in a regime of relatively small $N$. First, note that for increasing $N$, the total resource cost by $N = 10$ is already down to within 10% of the minimum resource cost reached at much larger $N$. Second, note that the theory is derived under the 'diffusive' memory storage assumption: that within a storage network, information loss is diffusive. Thus, the assumption implicitly made while varying the parameter $N$ in *Figure 4C* is that as the number of networks ($N$) is increased, the diffusivity $\mathcal{D} \propto 1/M$ per network will simply increase in proportion to keep $NM$ fixed. However, the dynamics of persistent activity networks do not remain purely diffusive once the resource per network drops below a certain level: a new kind of non-diffusive error can start to become important (Schwab DJ & Fiete I (in preparation)). In this regime, the effective diffusivity in the network can grow much faster than the inverse network size. The non-diffusive errors produce large, non-local errors (which may be consistent with 'pure guessing' or 'sudden death' errors sometimes reported in memory psychophysics [*Zhang and Luck, 2009*]). It is possible that the memory networks operate in a regime where each channel (memory network) is allocated enough resources to mostly avoid non-diffusive errors, and this limits the number of networks.

## Discussion

### Key contributions

We have provided a fundamental lower-bound on the error of recall in short-term memory as a function of item number and storage duration, if information is stored in graded persistent activity networks (our noisy channels). This bound on performance with an underlying graded persistent activity mechanism provides a reference point for comparison with human performance regardless of whether the brain employs strong encoding and decoding processes in its memory systems. The comparison can yield insights into the strategies the brain does employ.

Next, we used empirical data from analog measurements of memory error as a function of both temporal delay and the number of stored items. Using results from the theory of diffusion on continuous attractor manifolds in neural networks, we derived an expression for memory performance if the memorized variables were stored directly in graded persistent activity networks. The resulting predictions did not match human performance. The mismatch invites further investigation into whether and how direct-storage models can be modified to account for real memory performance.

Finally, we found that the bound from theory provided an (unexpectedly) good match to human performance, *Figure 4*. We are not privy to the actual values of the parameters $N, 1/2\mathcal{D}$ in the brain and it is possible the brain uses a value of, to take an arbitrary example, $\approx 5 \times N$ to achieve a performance reached with $N$ in *Equation 2*, which would be (quantitatively) 'suboptimal'. Nevertheless, the possibility that the brain might perform qualitatively according to the functional form of the theoretical bound is highly nontrivial: As we have seen, the addition of appropriate encoding and

decoding systems can reduce the degradation in accuracy from scaling polynomially ($\sim 1/N$) in the number of neurons, as in direct storage, to scaling exponentially ($\sim e^{-\alpha N}$ for some $\alpha > 0$). This is a startling possibility that requires more rigorous examination in future work.

## Are neural representations consistent with exponentially strong codes?

Typical population codes for analog variables, as presently understood, exhibit linear gains in performance with $N$; such codes involve neurons with single-bump or ramp-like tuning curves that are offset or scaled copies of one another. For related reasons, persistent activity networks with such tuning curves also exhibit linear gains in memory performance with $N$ (*Burak and Fiete, 2012*). These 'classical population codes' are ubiquitous in the sensory and motor peripheries as well as some cognitive areas. So far, the only example of an analog neural code known in principle to be capable of exponential scaling with $N$ is the periodic, multi-scale code for location in grid cells of the mammalian entorhinal cortex (*Hafting et al., 2005*; *Sreenivasan and Fiete, 2011*; *Mathis et al., 2012*) : with this code, animals can represent an exponentially large set of distinct locations at a fixed local spatial resolution using linearly many neurons (*Fiete et al., 2008*; *Sreenivasan and Fiete, 2011*).

A literal analogy with grid cells would imply that all such codes should look periodic as a function of the represented variable, with a range of periods. A more general view is that the exponential capacity of the grid cell code results from two related features: First, no one group of grid cells with a common spatial tuning period carries full information about the coded variable (the spatial location of the animal) – location cannot be uniquely specified by the spatially periodic group response even in the absence of any noise. Second, the partial location information in different groups is independent because of the distinct spatial periods across groups (*Sreenivasan and Fiete, 2011*). In this more general view, strong codes need not be periodic, but there should be multiple populations that encode different, independent 'parts' of the same variable, which would be manifest as different sub-populations with diverse tuning profiles, and mixed selectivity to multiple variables.

It remains to be seen whether neural representations for short-term visual memory are consistent with strong codes. Intriguingly, neural responses for short-term memory are diverse and do not exhibit tuning that is as simple or uniform as typical for classical population codes (*Miller et al., 1996*; *Fuster and Alexander, 1971*; *Romo et al., 1999*; *Wang, 2001*; *Funahashi, 2006*; *Fuster and Jervey, 1981*; *Rigotti et al., 2013*). An interesting prediction of the well-coded model, amenable to experimental testing, is that the representation within a memory channel must be in an optimized format, and that this format is not necessarily the same format that information was initially presented in. The brain would have to perform a transformation from stimulus-space into a well-coded form, and one might expect to observe this transition of the representation at encoding. (See, e.g., recent works (*Murray et al., 2017*; *Spaak et al., 2017*), which show the existence of complex and heterogeneous dynamic transformations in primate prefrontal cortex during working memory tasks.) The less orthogonal the original stimulus space is to noise during storage and the more optimized the code for storage to resist degradation, the more different the mnemonic code will be from the sample-evoked signal. Studies that attempt to decode a stimulus from delay-period neural or BOLD activity on the basis of tuning curves obtained from the stimulus-evoked period are well-suited to test this question (*Zarahn et al., 1999*; *Courtney et al., 1997*; *Pessoa et al., 2002*; *Jha and McCarthy, 2000*; *Miller et al., 1996*; *Baeg et al., 2003*; *Meyers et al., 2008*; *Stokes et al., 2013*) : If it is possible to use early stimulus-evoked responses to accurately decode the stimulus over the delay-period (*Zarahn et al., 1999*; *Courtney et al., 1997*; *Pessoa et al., 2002*; *Jha and McCarthy, 2000*; *Miller et al., 1996*), it would suggest that information is not re-coded for noise resistance. On the other hand, a representation that is reshaped during the delay period relative to the stimulus-evoked response (*Baeg et al., 2003*; *Meyers et al., 2008*; *Stokes et al., 2013*) might support the possibility of re-coding for storage.

On the other hand, the encoding and decoding steps for strong codes add considerable complexity to the storage task, and it is unclear whether these steps can be performed efficiently so that the efficiencies of these codes are not nullified by their costs. In light of our current results, it will be interesting to further probe with neurophysiological tools whether storage for short-term visual memory is consistent with strong neural codes. With psychophysics, it will be important to compare human performance and the information-theoretic bound in greater detail. On the theoretical side, studying the decoding complexity of exponential neural codes is a topic of ongoing work

(*Fiete et al., 2014*; *Chaudhuri and Fiete, 2015*), where we find that non-sparse codes made up of a product of many constraints on small subsets of the codewords might be amenable to strong error correction through simple neural dynamics.

## Relationship to existing work and questions for the future

Compared to other information-theoretic considerations of memory (*Brady et al., 2009*; *Sims et al., 2012*), the distinguishing feature of our approach is our focus on neuron- or circuit-level noise and the fundamental limits such noise will impose on persistence.

Our theoretical framework permits the incorporation of many additional elements: Variable allocation of resources during stimulus presentation based on task complexity, perceived importance, attention, and information loading rate, may all be incorporated into the present framework. This can be achieved by modeling $1/2\mathcal{D}$ and $N$ as dependent functions (e.g. as done in [*van den Berg et al., 2012*; *Sims, 2012*; *Elmore et al., 2011*]) rather than independent parameters, and by exploiting the flexibility allowed by our model in uneven resource allocation across items in the display (Materials and methods).

The memory psychophysics literature contains evidence of more complex memory effects, including a type of response called 'sudden death' or pure guessing (*Zhang and Luck, 2009*; *Anderson et al., 2011*). These responses are characterized by not being localized around the true value of the cued variable, and contribute a uniform or pedestal component to the response distribution. Other studies show that these apparent pedestals may not be a separate phenomenon and can, at least in some cases, be modeled by a simple growth in the variance over a bounded (circular) variable of a unimodal response distribution that remains centered at the cue location (*van den Berg et al., 2012*; *Bays, 2014*; *Ma et al., 2014*). In our framework, good encoding ensures that for noise below a threshold, the decoder can recover an improved estimate of the stored variable; however, strong codes exhibit sharp threshold behavior as the noise in the channel is varied smoothly. Once the noise per channel grows beyond the threshold, so-called catastrophic or threshold errors will occur, and the errors will become non-local: this phenomenon will look like sudden death in the memory report. In this sense, an optimal coding and decoding framework operating on top of continuously diffusing states in memory networks is consistent with the existence of sudden death or pure guessing-like responses, even without a distinct underlying mechanistic process in the memory networks themselves. We note, however, that the fits to the data shown here were all in the below-threshold regime.

Another complex effect in memory psychophysics is misbinding, in which one or more of the multiple features (color, orientation, size, etc.) of an item are mistakenly associated with those from another item. This work should be viewed as a model of single-feature memory. Very recently, there have been attempts to model misbinding (*Matthey et al., 2015*). It may be possible to extend the present model in the direction of (*Matthey et al., 2015*) by imagining the memory networks to be multi-dimensional attractors encoding multiple features of an item.

It will be important to understand whether in the direct coding model, modifications with plausible biological interpretations can lead to significantly better agreement with the data. From a purely curve-fitting perspective, the model requires stronger-than-linear improvement in recall accuracy with declining item number, and one might thus convert the combined resource parameter $N/\mathcal{D}$ in *Equation 1* into a function that varies inversely with $K$. This step would result in a better fit, but would correspond in the direct storage model to an *increased* allocation of total memory resources when the task involves *fewer* items, an implausible modification. Alternatively, if multiple items are stored within a single persistent activity network, collision effects can limit performance for larger item numbers (*Wei et al., 2012*), but a quantitative result on performance as a function of delay time and item number remain to be worked out. Further examination of the types of data we have considered here, with respect to predictions that would result from a memory model dependent on direct storage of variables into persistent activity network(s), should help further the goal of linking short-term memory performance with neural network models of persistent activity.

Finally, note that our results stem from considering a specific hypothesis about the neural substrates of short-term memory (that memory is stored in a continuum of persistent activity states) and from the assumption that forgetting in short-term memory is undesirable but neural resources required to maintain information have a cost. It will also be interesting to consider the possibility of information storage in discrete rather than graded persistent activity states, with appropriate

discretization of analog information before storage. Such storage networks will yield different bounds on memory performance than derived here (*Koulakov et al., 2002*; *Goldman et al., 2003*; *Fiete et al., 2014*), which should include the existence of small analog errors arising from discretization at the encoding stage, with little degradation over time because of the resistance of discrete states to noise. Also of great interest is to obtain predictions about degradation of short-term memory in activity-silent mechanisms such as synaptic facilitation (*Barak and Tsodyks, 2014*; *Mi et al., 2017*; *Stokes, 2015*; *Lundqvist et al., 2016*). A distinct alternate perspective on the limited persistence of short-term memory is that forgetting is a design feature that continually clears the memory buffer for future use and that limited memory allows for optimal search and computation that favors generalization instead of overfitting (*Cowan, 2001*). In this view, neural noise and resource constraints are not bottlenecks and there may be little imperative to optimize neural codes for greater persistence and capacity. To this end, it will be interesting to consider predictions from a theory in which limited memory is a feature, against the predictions we have presented here from the perspective that the neural system must work to avoid forgetting.

## Materials and methods

### Human psychophysics experiments

Ten neurologically normal subjects (age range 19-35 yr) participated in the experiment after giving informed consent. All subjects reported normal or corrected-to-normal visual acuity. Stimuli were presented at a viewing distance of 60 cm on a 21" CRT monitor. Each trial began with the presentation of a central fixation cross (white, $0.8°$ diameter) for 500 milliseconds, followed by a memory array consisted of 1, 2, 4, or 6 oriented bars ($2°$ of visual angle) presented on a grey background on an imaginary circle (radius $4.4°$) around fixation with equal inter-item distances (centre to centre). The colors of the bars in each trial were randomly selected out of eight easily-distinguishable colors. The stimulus display was followed by a blank delay of $0.1, 1, 2$ or $3$ seconds and at the end of each sequence, recall for one of the items was tested by displaying a 'probe' bar of the same color with a random orientation. Subjects were instructed to rotate the probe using a response dial (Logitech Intl. SA) to match the remembered orientation of the item of the same color in the sequence - henceforth termed the target. Each of the participants performed between 11 and 15 blocks of 80 trials. Each block consisted of 20 trials for each of the 4 possible item numbers, consisting of 5 trials for each delay duration.

### Overview of theoretical framework and key steps

#### Channel coding and channel rate

Consider transmitting information about $K$ scalar variables in the form of codewords of power 1 (i.e., $\sum_{k=1}^{K} P^{(k)} = 1$, where $P^{(k)}$ is the average power allocated to encode item $k$, with the average taken over $N$ different channel uses, so that the average power actually used is $\frac{1}{N}\sum_{i=1}^{N}(X_i^{(k)})^2 \le P^{(k)}$. The number of channel uses, $N$, is equivalent in our memory framework to the number of parallel memory channels, each of which introduces a Gaussian white noise of variance $2\mathcal{D}T$. The rate of growth of variance of the variable stored in persistent activity networks, $2\mathcal{D}$, is derived in *Burak and Fiete (2012)*; here, when we refer to this diffusivity, it is in dimensionless units where the variable is normalized by its range.

The information throughput (i.e., the information rate per channel use, also known as channel rate) for such channels is bounded by (see Appendix for details):

$$R^{\mathcal{S}}(T) \equiv \sum_{k \in \mathcal{S}} R^{(k)} \le \frac{1}{2}\log\left(1 + \frac{\sum_{k \in \mathcal{S}} P^{(k)}}{2\mathcal{D}T}\right) \tag{3}$$

where $\mathcal{S}$ refers to any subset of the the $K$ items, $\{1, \cdots, K\}$. *Equation 3* defines an entire region of information rates that are achievable: the total encoding power or the total channel rate, or both, may be allocated to a single item, or distributed across multiple items. Thus, the expression of *Equation 3* is compatible with interpretations of memory as either a continuous or a discrete resource (*van den Berg et al., 2012*; *Zhang and Luck, 2008*). (E.g., setting $P^{(k)} = 0$ for any $k \ge 5$, would

correspond to a 4-slot conceptualization of short-term memory. Distributing $P^{(k)} = 1/K$ for any variable number $K$ of statistically similar items, would more closely describe a continuous resource model.) For both conceptualizations, this framework would allow us to consider, if the experiment setup warranted, different allocations of power $P^{(k)}$ and information rates across the encoded items.

For the delayed orientation matching task considered here, all presented items have equal complexity and *a priori* importance, so the relevant case is $P^{(k)} = 1/K$ for all $k = 1, \cdots, K$, together with equal-rate allocation, $R^{(1)} = \cdots = R^{(K)}$, resulting in the following bound on per-item or per-feature information throughput in the noisy channel (see Appendix for more detail):

$$R^{(k)}(T) \leq \frac{1}{2K} \log\left(1 + \frac{1}{2\mathcal{D}T}\right). \tag{4}$$

Next we consider how this bound on information rate in turn constrains the reconstruction error of the source variable (i.e., the $K$-variable vector to be memorized, $\vec{\phi}$).

## Source coding and rate-distortion theory

At a source coder that compresses a source variable, rate-distortion theory relates the source rate to the distortion in reconstructing the source, at least for specific source distributions and specific error (distortion) metrics. For instance, if the source variables are each drawn uniformly from the interval $[0, \Phi]$, then the mean-squared error in reconstructing the source, $D_{\mathrm{MSE}}$, is related to the source rate $R$ through the rate-distortion function (see Appendix):

$$\frac{1}{2} \log\left(\frac{\Phi^2}{2\pi e D_{\mathrm{MSE}}}\right) \leq R \leq \frac{1}{2} \log\left(\frac{\Phi^2}{12 D_{\mathrm{MSE}}}\right). \tag{5}$$

## Joint source-channel coding

If the source rate is set to equal the maximal channel rate of *Equation 4*, then use the expression of *Equation 5* from rate-distortion theory, we obtain the predicted bound on distortion in the source variable after source coding and channel transmission. This predicted distortion bound is given in *Equation 2*. In general problems of information transmission through an noisy channel, it is not necessarily jointly optimal to separately derive the optimal channel rate and the optimal distortion for a given source rate, and then to set the source rate to equal the maximal channel rate; the total distortion of the source passed through the channel need not be lower-bounded by the resulting expression. However, in our case of interest the two-step procedure described above, deriving first the channel capacity then inserting the capacity into the rate-distortion equation, yields a tight bound on distortion for the memory framework.

This concludes the basic derivation, in outline form, of the main theoretical result of the manuscript. The Supplementary Information supplies more steps and detail.

## Fitting of theory to data

In all fits of theory to data (for direct and well-coded storage), we assume that recall error at the shortest storage interval of 100 ms reflects *baseline errors* unrelated to the temporal loss of recall accuracy from noisy storage that is the focus of the present work. Under the assumption that this early ('initial') error is independent of the additional errors accrued over the storage period, it is appropriate to treat the baseline ($T = 100$ ms) MSE as an additive contribution to the rest of the MSE (the variance of the sum of independent random variables is the sum of their variances). For this reason, we are justified in treating the $T = 100$ ms errors as given by the data and setting these points as the initial offsets of the theory curves, which go on to explain the temporal (item-dependent) degradation of information placed in noisy storage.

The curves are fit by minimizing the summed weighted squared error of the theoretical prediction in fitting the subject-averaged performance data over all item numbers and storage durations. The theoretical predictions are given by *Equation 1* for direct storage and *Equation 2* for well-coded storage. The weights in the weighted least-squares are the inverse SEMs for each (item, storage duration) pair. The parameters of the fit are $N/2\mathcal{D}$ (direct storage model) or $N$ and $2\mathcal{D}$ (well-coded model). The parameter value selected is common across all item numbers and storage durations.

The *p* values given in the main paper quantify how likely the data means are to have been based on samples from a Gaussian distribution centered on the theoretical prediction.

## Model comparison with the bayesian information criterion

The Bayesian Information Criterion (BIC) is a likelihood-based method for model comparison, with a penalty term that takes into account the number of parameters used in the candidate models. BIC is a Bayesian model comparison method, as discussed in *Kass and Raftery (1995)*

Given data $x$ that are (assumed to be) drawn from a distribution in the exponential family and a model $M(\vec{\theta})$ with associated parameters $\vec{\theta}$ ($\vec{\theta}$ is a vector of $k$ parameters), the BIC is given by:

$$\mathrm{BIC} = -2\hat{L} + k\ln(2\pi n) \tag{6}$$

where $n$ is the number of observations, and $\hat{L}$ is the likelihood of the model (with parameters $\vec{\theta}$ selected by maximum likelihood). The smaller the BIC, the better the model. The more positive the difference

$$\Delta\mathrm{BIC} = BIC(M_2) - BIC(M_1) \tag{7}$$

between a pair of models $M_1(\vec{\theta}_1)$ and $M_2(\vec{\theta}_2)$ (with associated parameters $\vec{\theta}_1, \vec{\theta}_2$, respectively, possibly of different dimensions $k_1 \neq k_2$), the stronger the evidence for $M_1$.

To obtain the BIC for the direct and coded models, the model distributions are taken to be Gaussians whose means (for each item and delay) are given by the theoretical results of *Equations 1 and 2*, respectively, and whose variance is given by the empirically measured data variance across trials and subjects, computed separately per item and delay. We used the parameters $N = 10, 1/2\mathcal{D} = 2.28$ for the well-coded storage model, and $(2\mathcal{D}/N) = 3.24 \times 10^{-7}$ for the direct storage model, to obtain $\Delta\mathrm{BIC} = 172.67$. The empirical response variance is computed over each trial for each subjects, for a total of $n = 660$ observations for each $(T, K)$ or (delay interval, item number) pair. The number of parameters is $k = 1$ for direct storage and $k = 2$ for well-coded storage. Setting the parameter numbers to $k = 1 + 4$ and $k = 2 + 4$ to take into account the 4 values of response errors at the shortest delay at $T = 100$ ms does not change the $\Delta\mathrm{BIC}$ score because the score is dominated by the likelihood term, so that these changes in the parameter penalty term have negligible effect.

## Additional information

### Funding

| Funder | Grant reference number | Author |
|---|---|---|
| National Science Foundation | IIS-1464349 | Onur Ozan Koyluoglu |
| Israel Science Foundation | 1747/14 | Yoni Pertzov |
| MRC Clinician Scientist Fellowship | MR/P00878X | Sanjay Manohar |
| National Institute for Health Research | Oxford Biomedical Centre | Masud Husain |
| Wellcome Trust | | Masud Husain |
| National Science Foundation | IIS-1148973 | Ila R Fiete |
| Simons Foundation | | Ila R Fiete |
| Howard Hughes Medical Institute | Faculty Scholar Award | Ila R Fiete |

The funders had no role in study design, data collection and interpretation, or the decision to submit the work for publication.

### Author contributions

Onur Ozan Koyluoglu, Conceptualization, Software, Formal analysis, Writing—original draft, Writing—review and editing; Yoni Pertzov, Data curation, Software, Writing—review and editing; Sanjay

Manohar, Masud Husain, Data curation, Writing—review and editing; Ila R Fiete, Conceptualization, Software, Writing—original draft, Writing—review and editing

## Author ORCIDs
Onur Ozan Koyluoglu (iD) http://orcid.org/0000-0001-8512-4755
Sanjay Manohar (iD) http://orcid.org/0000-0003-0735-4349

## Ethics
Human subjects: The study reported here conform to the Declaration of Helsinki and all procedures were approved by the ethics committee of the National Hospital for Neurology and Neurosurgery (NHNN) prior to the study commencing. Research Ethics Committee number (ERC) 04/Q0406/60. Personal information about individuals was password protected and saved in compliance to the Data Protection Act 1998 (DPA).

## Decision letter and Author response
Decision letter https://doi.org/10.7554/eLife.22225.016
Author response https://doi.org/10.7554/eLife.22225.017

# Additional files
## Supplementary files
• Transparent reporting form
DOI: https://doi.org/10.7554/eLife.22225.014

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

## Appendix

DOI: https://doi.org/10.7554/eLife.22225.015

# Joint source-channel coding and memory: justification and main results

## Noisy information channels as a component of short-term memory systems

Noisy information channels have traditionally been used to model communication systems: in satellite or cell-phone communications, the transmitted information is degraded during passage from one point to another (*Shannon, 1959*; *Wang, 2001*; *Cover and Thomas, 1991*). Such transmission and degradation over space is referred to as a channel use. However, noisy channels are apt descriptors of any system in which information is put in to be accessed at a different place or a different time, with loss occurring in-between (*Shannon, 1959*; *Wang, 2001*; *Cover and Thomas, 1991*). Thus, hard drives are channels, with the main channel noise being the probability of random bit flips (from high-energy cosmic rays). Similarly, neural short-term memory systems store information and are subject to unavoidable loss because of the stochasticity of neural spiking and synaptic activation. In this sense, noise-induced loss in persistent activity networks is like passing the stored information through a noisy channel.

## Channel coding

In channel coding, a message is first encoded to add redundancy, then transmitted through the noisy channel, and finally decoded at the decoder. Here, we establish the terminology and basic results from Shannon's noisy channel coding theory (*Shannon, 1959*; *Cover and Thomas, 1991*), which are used in the main paper.

First, consider a task that involves storing or communicating a simple message, $q$, where $q$ is a uniformly distributed index taking one of $Q$ values: $q \in \{1, \cdots, Q\}$. The message $q$ is encoded according to a deterministic vector function (an encoding function), to generate the $N$-dimensional vector $\mathbf{x}(q) = (x_1(q), x_2(q), \cdots, x_N(q))$, **Figure 1**. This is the *channel-coding* step. The codeword $\mathbf{x}(q)$, is redundant, is sent through the noisy channel, which produces an output $\mathbf{y}$ according to some conditional distribution $p(\mathbf{y}|\mathbf{x})$ ($\mathbf{y}$ is an $N$-dimensional vector; the channel is specified by the distribution $p(\mathbf{y}|\mathbf{x})$). In a memoryless channel (no feedback from the decoder at the end of the channel back to the encoder at the mouth of the channel), the channel obeys

$$p(\mathbf{y}|\mathbf{x}) = \prod_{n=1}^{N} p(y_n|x_n), \tag{8}$$

where all distributions $p(y_n|x_n)$ represent an identical distribution that defines the channel (*Cover and Thomas, 1991*). In this setup, transmission of the scalar source variable $q$ involves $N$ independent channel uses.

The decoder constructs a mapping $\mathbf{y} \rightarrow \{1, \cdots, Q\}$, to make an estimate $\hat{q}$ of the received message from the channel outputs $\mathbf{y}$. If $\hat{q} \neq q$, the decoder has made an error. The error probability is the probability that $q$ is decoded incorrectly, averaged over all $q$. This scenario, in which $q$, which is a single number (and represents one of the messages to be communicated) and the decoder receives a single number (observation) from each channel use, is referred to as point-to-point communication (*Cover and Thomas, 1991*).

If the decoder can correctly decode $q$, the channel communication rate (also known as the rate per channel use), which quantifies how many information bits (about $q$) are transmitted per entry of the coded message $\mathbf{x}$, is given by $R = \log_2(Q)/N$. Shannon showed in his noisy channel coding theorem (*Shannon, 1959*; *Cover and Thomas, 1991*) that for any channel, in

the limit $N \to \infty$, it is possible in principle to communicate error-free through the channel at any rate up to the *channel capacity C*, defined by:

$$C = \max_{p(x)} \frac{1}{N} I(\mathbf{x}; \mathbf{y}).$$

(9)

For specific channels, it is possible to explicitly compute the channel capacity in terms of interesting parameters of the channel model and encoder; below, we will state such results for our channels of interest, for subsequent use in our theoretical analysis.

### Point-to-point Gaussian channel with a power constraint

For a scalar quantity transmitted over an additive Gaussian white noise channel of variance $2\mathcal{D}T$, with an average power constraint $P$ for representing the codewords (i.e., $\frac{1}{N}\sum_{i=1}^{N} ||x_i||^2 \leq P$), the *channel capacity* , or maximum rate at which information can be transmitted without error, is given by (**Cover and Thomas, 1991**) :

$$C = \frac{1}{2}\log\left(1 + \frac{P}{2\mathcal{D}T}\right).$$

(10)

### Gaussian multiple-access channel

Next, suppose the message is itself multi-dimensional (of dimension $K$), so that the message is $\mathbf{q} = (q^1, \cdots, q^K)$. (In a memory task, these $K$ variables may correspond to different features of one item, or one feature each of multiple items, or some distribution of features and items. All features of all items are simply considered as elements of the message, appropriately ordered.)

The general framework for such a scenario is the multiple-access channel (MAC). In a MAC, separate encoders each encode one message element $q^k$ ($k = 1, \cdots, K$), as an $N$-dimensional codeword $\mathbf{x}^k(q^k)$. The full message $\mathbf{q}$ is thus represented by a set of $K$ different $N$-dimensional codewords, $\mathbf{X}(\mathbf{q}) = (\mathbf{x}^1(q^1), \cdots, \mathbf{x}^K(q^K))$. The power of each encoder is limited to $P^{(k)}$ with a constraint on the summed power (we assume $\sum_{k=1}^{K} P^{(k)} \leq 1$.) The encoded outputs are transmitted through a channel with a single receiver at the end.

As before, we consider the channel to be Gaussian. In this Gaussian MAC model, the channel output $\mathbf{y}$ is a single $N$-dimensional vector, like the output in the point-to-point communication case (**Cover and Thomas, 1991**). The MAC channel is defined by the distribution $p(\mathbf{y}|\mathbf{X}) = p(\mathbf{y}|\mathbf{x}^1, \cdots, \mathbf{x}^K)$. For a Gaussian MAC, $p(\mathbf{y}|\mathbf{X})$ is a Gaussian distribution with mean equal to $\sum_{k=1}^{K} \mathbf{x}^k$ and variance equal to the noise variance. The decoder is tasked with reconstructing all $K$ elements of $\mathbf{q}$ from the $N$-dimensional $\mathbf{y}$.

The probability of error is defined as the average probability of error across all $K$ entries of the message. The fundamental limit on information transmission over the MAC is not a single number, but a region in a $K$-dimensional space: It is possible to allocate power and thus rates differentially to different entries of the message $\mathbf{q}$, and information capacity varies based on allocation. Through Shannon's channel coding theorem, the region of achievable information rates for the Gaussian MAC with noise variance $2\mathcal{D}T$ is given by:

$$R^{\mathcal{S}} \leq \frac{1}{2}\log\left(1 + \frac{\sum_{k \in \mathcal{S}} P^{(k)}}{2\mathcal{D}T}\right),$$

(11)

where $\mathcal{S}$ refers to any subset of $\{1, \cdots, K\}$, and we represent the summed rate for a given $\mathcal{S}$ as $R^{\mathcal{S}} = \sum_{k \in \mathcal{S}} R^{(k)}$. In memory tasks, we assume the total power constraint is constant, regardless of the number of items, and $K$ corresponds to the number of items. Thus, power allocation per item will generally vary (decrease) with item number.

To summarize, we have a fundamental limit on information transmission rates in a Gaussian multiple-access channel as described above.

## Capacity of a Gaussian MAC with equal per-item rate equals point-to-point channel capacity

The summed information rate through a Gaussian MAC channel is maximized when the per-item rate is equal across items. Moreover, at this equal-rate per-item point, the Gaussian MAC model corresponds directly to a point-to-point Gaussian (AWGN) channel coding model, where the channel input has an average power constraint $P$, which is set to $P = \sum_k P^{(k)}$, where $P^{(k)}$ is the power constraint on the channel input of the $k$-th encoder of the original Gaussian MAC model. In this equivalent AWGN model, a single encoder is responsible for transmitting all of the $K$ message elements, by dividing the point-to-point channel capacity equally among the message elements. The maximum information rate in a point-to-point AWGN channel is $(1/2)\log(1 + \mathrm{SNR})$, and therefore the information rate per item, if the rate is divided evenly over all $K$ items, is $R^{(k)} = (1/2K)\log(1 + \mathrm{SNR})$. This capacity can be achieved by setting the inputs for the AWGN point-to-point channel to be the $N$-dimensional vector $\mathbf{x}$, with

$$\mathbf{x} = \sum_{k=1}^{K} \mathbf{x}^k(q^k), \text{ where } \mathbf{x}^k(q^k) \text{ are the set of } K \text{ vectors of length } N \text{ generated from the encoders}$$

of the Gaussian MAC. The $i$th component $x_i$ of $\mathbf{x}$ is $x_i = \sum_{k=1}^{K} x_i^k(q^k)$, where $x_i^k(q^k)$ is the $i$th

element of the vector $\mathbf{x}^k$ which encodes the message element $q^k$, and therefore $x_i$ contains information about all components of the message (joint representation of message elements).

Comparing the expression for the Gaussian MAC information rate with the capacity result from the corresponding point-to-point Gaussian channel, $R^{(k)} = (1/2K)\log(1 + \mathrm{SNR})$, it is clear that the summed rate of the equal-rate per-item Gaussian MAC can achieve the same (optimal) information rate per item as the point-to-point AWGN channel.

*Figure 4B* of our main manuscript may be viewed as depicting the AWGN point-to-point channel, with a scalar input $x_i$ to each of the $N$ memory networks (AWGN channels). It is interesting to note that both the AWGN channel and Gaussian MAC models suggest that the brain might encode distinct items independently but then store them jointly.

## Point-to-point communication through a Gaussian channel with a peak amplitude constraint

Suppose the codewords are amplitude-limited, rather than collectively power-limited, so that each element $||x_i|| \leq A$ for some amplitude $A$. If we are considering each entry of the codeword as being stored in a persistent activity network, then the maximal range of each codeword entry is constrained, rather than just the average power across entries. In this sense, amplitude-constrained channels may be more apt descriptors than power-constrained channels.

For comparison with the capacity of a Gaussian channel with a power constraint $P$, we set without loss of generality $A = \sqrt{P}$. Then, for a scalar quantity transmitted with this amplitude constraint over an additive Gaussian white noise channel of variance $2\mathcal{D}T$, the *channel capacity* is similar to that of the power-constrained Gaussian channel, but with the cost of a modest multiplicative pre-factor $c$ that is smaller than, but close to size 1 (*Softky and Koch, 1993*; *Raginsky, 2008*):

$$C = \frac{c}{2}\log\left(1 + \frac{P}{2\mathcal{D}T}\right). \tag{12}$$

If the SNR ($= \frac{P}{2\mathcal{D}T}$) is such that $\sqrt{\mathrm{SNR}} < 1.05$, then $c \in [0.8, 1]$ (*Raginsky, 2008*). Therefore, channel capacity of the amplitude-constrained Gaussian channel can be 80% or more of the channel capacity of the corresponding power-constrained Gaussian channel. In any case, the power-constrained Gaussian channel capacity expression is a good upper bound on the capacity of the amplitude-constrained version of that channel.

## Joint source-channel coding

In memory experiments, it is not possible to directly measure information throughput in the internal storage networks. Rather, a related quantity that can be measured, and is thus the quantity of interest, is the accuracy of recall. In this section, we describe how the general bound on information throughput in the storage networks – derived in the previous section – can be used to strictly upper-bound the accuracy of recall in a specific class of memory tasks.

Consider a task that involves storing or communicating a variable $\phi$. This variable is known as the information source. The information source may be analog or discrete, and uniform or not. To remove redundancies in the source distribution or to possibly even further compress the inputs (at the loss of information), the source may be passed through a source-coding step. (For instance, the real interval $[-1, 1]$ can be compressed through binary quantization into one bit by assigning the subinterval $[-1, 0]$ to the point 0, and $[0, 1]$ to 1, at the expense of precision.) The output of the source coder is known as the *message*, which was the assumed input to the noisy channel in the sections discussed above. The message is a uniformly distributed index $q$, taking one of $Q$ values, $q \in \{1, \cdots, Q\}$. The source rate is the number of bits allocated per source symbol, or $\log_2(Q)$.

For discrete, memoryless point-to-point Gaussian channels, Shannon's separation theorem (**Shannon, 1959**; **Cover and Thomas, 1991**) holds, which means that to obtain minimal distortion of a source variable that must be communicated through a noisy channel, it is optimal to separately compute the channel information rate, then set the source rate to equal the channel rate. Rate-distortion theory from source coding will then specify the lower bound on distortion with this scheme. Because the separation theorem holds for the point-to-point AWGN channel considered above, and because the point-to-point AWGN rate equals the maximal summed MAC rate, we can apply the separation theorem to our memory framework and then use rate-distortion theory to compute the lower bound on distortion.

To minimize distortion according to the separation theorem, we therefore set the source rate $\log_2(Q)$ to equal the maximum number of bits that may be transmitted error-free over the channel. With this choice, all messages are transmitted without error in the channel. Then, we apply rate-distortion theory to determine the minimum distortion achievable for the allocated source rate. For a given source rate allocation, the distortion depends on several factors: the statistics of the source (e.g. whether it is uniform, Gaussian, etc.), the source coding scheme, and on the distortion measure (e.g. mean absolute error (an L-1 norm), mean squared error (an L-2 norm), or another metric that quantifies the difference between the true source and its estimate). Closed-form expressions for minimum achievable distortion do not exist for arbitrary sources and distortion metrics, but crucially, there are some useful bounds on specific distortion measures including the mean squared error, which is our focus.

## Mean squared error (MSE) distortion

For arbitrary source distributions, the relationship between source rate ($R$ bits per source symbol) and minimum MSE distortion ($D_{\mathrm{MSE}}(R)$) at that rate, is given by:

$$h(\phi) - \frac{1}{2}\log(2\pi e D_{\mathrm{MSE}}(R)) \leq R \leq \frac{1}{2}\log\left(\frac{\sigma_\phi^2}{D_{\mathrm{MSE}}(R)}\right)$$

where $h(\phi)$ is the differential entropy of the source, $\sigma_\phi^2$ is the variance of the source, and $\log$ is in base-2. The inequality on the right is saturated (becomes an equality) for a Gaussian source (**Cover and Thomas, 1991**). The inequality on the left is the Shannon Lower Bound (**Sims et al., 2012**) on MSE distortion for arbitrary memoryless sources, and it, too, is saturated for a Gaussian source (**Cover and Thomas, 1991**).

Specializing the above expression to a uniform source over the interval $[0, \Phi]$, we have $h(\phi) = \log(\Phi)$, and $\sigma_\phi^2 = \Phi^2/12$. Thus, we obtain

$$\frac{1}{2}\log\left(\frac{\Phi^2}{2\pi e D_{\mathrm{MSE}}}\right) \leq R \leq \frac{1}{2}\log\left(\frac{\Phi^2}{12 D_{\mathrm{MSE}}}\right). \tag{13}$$

Inverting the inequalities above to obtain bounds on the MSE distortion, we have

$$\frac{\Phi^2}{2\pi e}2^{-2R} \leq D_{\mathrm{MSE}}(R) \leq \frac{\Phi^2}{12}2^{-2R}. \tag{14}$$

Note that the upper and lower bounds are identical in form – proportional to $\Phi^2 2^{-2R}$ – up to a constant prefactor that lies between $[1/2\pi e, 1/12]$. Thus, the lower bound on distortion is given by

$$D_{\mathrm{MSE}}(R,\Phi) = \frac{\alpha_{\mathrm{MSE}}\Phi^2}{2\pi e}2^{-2R}, \tag{15}$$

where $\alpha_{MSE}$ is an unknown constant of size about 1, somewhere in the range $[1, 2\pi e/12]$.

Now, we set the information rate $R$ for the source (bits per source symbol) in the equation above, to match the the maximum rate for error-free transmission in the noisy storage information channel. The maximum number of bits that can be stored error-free is $N$ times the channel capacity given in **Equation 4**, because **Equation 4** represents the information capacity for each channel use, and each of the $N$ storage networks represents one channel use. Thus, we have $R = NR^{(k)}(T)$, where $R^{(k)}(T)$ is given in **Equation 4**, and the minimum MSE distortion is:

$$D_{\mathrm{MSE}}(\Phi,K,T) = \frac{\alpha_{\mathrm{MSE}}\Phi^2}{2\pi e}\left(1+\frac{P}{2\mathcal{D}T}\right)^{-N/K}. \tag{16}$$

Because we are interested in the lower-bound on error, we set $\alpha_{\mathrm{MSE}}$ to the lower bound of its range, $\alpha_{\mathrm{MSE}} = 1$, so that we obtain the expression given in the main paper (**Equation 2**):

$$D_{MSE}(\Phi,K,T) = \frac{\Phi^2}{2\pi e}\left(1+\frac{P}{2\mathcal{D}T}\right)^{-N/K}. \tag{17}$$

Indeed, any other choice of $\alpha_{MSE}$ within its range $[1, 2\pi e/12]$ does not qualitatively affect our subsequent results in the main paper.

To summarize, we derived the bound given in **Equation 16** by separately combining two different bounds - the lower-bound on achievable distortion at a source for a given source rate and the upper-bound on information throughput in a noisy information channel. This combination of the two separate bounds, where each bound did not take into account the statistics of the other process (the source bound was computed independently of the channel and the channel independently of the source), is in general sub-optimal. It is tight (optimal) in this case only because the uniform source and Gaussian channel obey the conditions of Shannon's separation theorem, also known as the joint source-channel coding theorem (**Cover and Thomas, 1991**; **Wang, 2001**; **MacKay, 2002**; **Shannon, 1959**; **Viterbi and Omura, 1979**).

## Bound on recall accuracy for amplitude-constrained channels

As noted in Section 2 of the Appendix, the power-constrained channel capacity is an upper bound for the amplitude-constrained channel capacity (amplitude $A = \sqrt{P}$). It follows that the lower-bound on distortion for power-constrained channels, **Equation 16**, is a lower-bound on the amplitude-constrained channel. Further, because the channel capacity of an amplitude-constrained Gaussian channel is of the same form as the capacity of a power-constrained Gaussian channel, with a prefactor $c$ that is close to 1, we easily see that the specific expression for MSE distortion is modified to be:

$$D_{MSE}(\Phi,K,T) = \frac{\alpha_{MSE}\Phi^2}{2\pi e}\left(1+\frac{P}{2\mathcal{D}T}\right)^{-cN/K}. \tag{18}$$

Because $N$ is a free parameter of the theory, we may simply renormalize $cN$ to equal $N$. Thus, the theoretical prediction obtained for a power-constrained channel is the same in functional form as that for an amplitude-constrained channel.

In comparing the theoretical prediction against the predictions of direct storage in persistent activity networks, however, we should take into account the factor $c$, noting that to

produce an effective value of $N$ requires $N/c$ many networks, which is greater than $N$ because $c<1$.

## Non-asymptotic considerations

Many of the numerical fits in the paper involve values of $N$ that are not large: $N$ is of order 10. When transmitting information with smaller $N$, the error-free information rate is lower (**Polyanskiy et al., 2010**), or conversely, if transmitting at rates close to capacity with smaller numbers of channel uses ($N$) there can be decoding errors. In deriving our bound on distortion from joint-source channel coding theory, we inserted the asymptotic value of information rate (the capacity) into the rate-distortion function and assumed that information transmission at that rate would be error-free. If errors occur, the resulting distortion will be higher. It is important to note that, even far from the asymptotic limit in $N$, the derived lower-bound on distortion in **Equation 16** remains a strict lower-bound; non-asymptotic effects can raise the overall error, not lower it.

Nevertheless, it is of interest to consider how distortion may be modified for values of $N$ that are not asymptotically large. One would write the total non-asymptotic MSE distortion ($D_{MSE}^{\sim asymp}$) as the sum of terms:

$$D_{MSE}^{\sim asymp} = D_{MSE}(1-p_e) + D_e p_e. \tag{19}$$

Here, $D_{MSE}$ is the error-free distortion bound derived above, $p_e$ is the probability of error in the non-asymptotic regime, and $D_e$ is the distortion in case of error. If an error resulted in total loss of information about the transmitted (coded) variable, $D_e$ would scale as $\Phi^2$, independent of $N$ or other parameters in the problem. The only dependence on $N$ would then enter through the probability of error, $p_e$. The probability of error vanishes exponentially with $N$ (**Polyanskiy et al., 2010**), and can be small even for relatively small values of $N$. The second term is in practice a small contributor to the MSE. Alternatively, one can ask how small $N$ can be and at how far below the asymptotic capacity to enable information transmission at or below a given error rate. Analytical and numerical results in **Polyanskiy et al. (2010)** show that at SNR values lower than the estimated SNR in the memory system model ($SNR = P/2\mathcal{D}T = 1/2\mathcal{D}T \approx 2.2$ dB at $T = 3$ sec and $SNR \approx 4$ dB at $T = 2$ sec; while Figure 6 in (**Polyanskiy et al., 2010**) has $SNR = 0$ dB and $p_e = 10^{-3}$), it is possible to remain within a factor of $1/3$ of the asymptotic information capacity with $N<10$. Thus, the non-asymptotic expectation is that the information transmission rate should be scaled down from the asymptotically achievable information rate (the capacity) by some factor $c$ (in this case, $c \sim 3$). Thus, through **Equation 15**, we see that the bound on distortion will remain the same as in **Equation 2** of the main manuscript, with the replacement of $N/K$ in the exponent by $N/cK$. In other words, the previous values of the fit parameter $N$ in the fits would actually correspond to $cN$. Thus, it actually takes $c$ times more resources (where $c$ scales slowly with $1/N$) to achieve a given level of performance non-asymptotically as asymptotically.

To summarize, the bound on distortion given in **Equation 16** is still a strict lower-bound on distortion in the regime where $N$ is not asymptotically large; moreover, the functional form of the bound can remain largely the same in the non-asymptotic regime because the error probability is small for modest $N$. In addition, it is possible to achieve a given low error probability at a fixed SNR by simply decreasing the information rate, which increases distortion in a way that is effectively the same as increasing the value of the free parameter $N$.

## Direct (uncoded) storage in persistent activity networks

Modeling short-term memory as direct storage of variables in persistent activity networks, produces results that are inconsistent with the data, as shown in the main paper. To obtain predictions for persistence and capacity through direct storage in persistent activity networks, first consider storing a single circular orientation variable, for a single bar in the delayed orientation matching task, as a bump in one ring network (**Ben-Yishai et al., 1995**; **Amit, 1992**; **Zhang, 1996**). The ring network would have neurons from all the $N$ storage networks in our short-term memory system pooled together, thus the network is $N$ times larger. The mean squared error of a variable stored in a continuous attractor neural network

with stochastic neural spiking grows linearly with the storage interval $T$ over short intervals (with 'short' defined as all intervals before the root-mean squared error has grown to be an appreciable fraction of the range of the variable, $2\pi$). Let $\phi/\Phi$ be the coded variable, with $\phi \in [0, \Phi]$. If the rate of growth of error in the individual storage networks of the main paper is $2\mathcal{D}$ (recall that $\bar{D} = D/P$, where $D$ is coefficient of diffusion (**Burak and Fiete, 2012**); thus, the quantity $\bar{D}$ describes the rate at which the stored variable drifts away from its initial value, normalized by the squared range of the variable, per unit power of the representation; alternatively, we may think of the total representional power as being normalized to 1 in all cases), then the rate of growth of squared error in the single ring network is $\sim 2\mathcal{D}/N$ (**Burak and Fiete, 2012**). The factor of $N$ enters because if all other quantities are held fixed, the diffusion coefficient in continuous attractor memory networks is inversely proportional to network size. Thus, the squared error in the variable at short times $T$ is given by $\langle (\phi(T) - \phi(0))^2 \rangle / \Phi^2 = 2\mathcal{D}T/N$. In other words, we have

$$D_{MSE}(\Phi, K = 1, T) = \Phi^2 \frac{2\mathcal{D}T}{N} \tag{20}$$

Next, consider storing $K$ scalar variables, with each component ranging in $[0, \Phi]$, and represented in one of $K$ different small networks, constructed from the single storage network above. Thus, its size is $1/K$ of the above. Relative to **Equation 20** above, we therefore have

$$D_{\mathrm{MSE}}(\Phi, K, T) = \Phi^2 \frac{2\mathcal{D}KT}{N} \tag{21}$$

In other words, for memory systems involving direct storage in persistent activity networks without special encoding, we expect the squared error to grow linearly with $K$ and $T$. The prediction of uncoded storage in persistent activity networks can be compared directly with the prediction from encoded storage (**Equation 2**), because they involve the same parameters and the same resource use in the memory networks. While adding a proper encoding stage can reduce storage errors exponentially in $N$, uncoded storage results in decreases with $N$ that are merely polynomial (more specifically, scaling as $N^{-1}$).

Finally, one may consider directly storing the $K$-dimensional variable in a single persistent activity network that is a $K$-dimensional ring network (a $K$-torus). In this situation, the neurons have to be arranged so the number of neurons per linear dimension of the network scales as $N^{1/K}$. Thus, the rate of growth of squared error along each dimension of the network scales as $2\mathcal{D}/(N^{1/K})$, and we have

$$D_{\mathrm{MSE}}(\Phi, K, T) = \Phi^2 \frac{2\mathcal{D}T}{N^{1/K}} \tag{22}$$

This scaling with $T$ remains linear, while the improvement in squared error with $N$ is weaker than the scaling in **Equation 21**, which in turn is weaker than the scaling in **Equation 2**, and consequently produces worse fits to the data than does **Equation 21**. Therefore, we have chosen to contrast the better of two scenarios of direct (uncoded) storage, **Equation 21**, against the predictions of the theory of short-term memory proposed in this work.

## Comparison of direct storage against coded storage in power- or amplitude-constrained channels

In the main text, we compared not only how the predictions of coded versus direct storage compare with each other as a function of $T$ and $K$, but also compared total resource use to achieve a given performance with the two different models of storage. In the latter comparison, we derive the total neural resource, $N/2\mathcal{D}$, required in the two schemes. We report that direct storage requires a $\sim 40$-fold larger $N/2\mathcal{D}$ than coded storage, basing our results on the expression for coded storage in power-constrained channels. As noted in Section 3 of the Appendix, the effective $N$ for an amplitude-constrained channel, which might be a more apt constraint for persistent activity networks with bounded ranges, is actually $N/c$, where $c$ is a prefactor close to but smaller than 1, that represents the fractional loss in channel

capacity incurred by enforcing an amplitude rather than power constraint. As described in (*Raginsky, 2008*) (see also related work in (*Softky and Koch, 1993*) ), the cost of replacing a power constraint by an amplitude constraint is modest, with $c \in [0.8, 1]$ for an appropriate regime of channel SNR (this is the regime of SNR for our fits to the data). Thus, even with an amplitude constraint for the coded memory scenario, direct storage would require a $\sim 30$-fold larger $N/2\mathcal{D}$.

## Performance of individual subjects and comparison with theory

Here, we supply the data from individual subjects, as well as fits of the theory of *Equation 2* and the direct storage model 1 to their performance.

The individual subject responses and the fits of the well-coded storage model are shown in *Figure 4—figure supplement 1*. We first plot the quality-of-fit or energy surface of the fits of the well-coded model to the individual subject data (top two rows in *Figure 4—figure supplement 1*), as the two parameters of the model are varied. These individual-subject solution spaces look qualitatively similar to the across-subject aggregates reported in the main manuscript. All subjects exhibit a 1D manifold of 'good' parameter settings, along which the model provides a reasonable match to the data. The quality of fit along the 1D manifold (valley) is shown in the next two rows of *Figure 4—figure supplement 1*; based on the local minima of these curves, we infer the optimal settings of $N$ and $1/2\mathcal{D}$ for each subject. The differences between individuals emerges in that the best $N$ values range between 2 and 20, and that for most subjects, the best values range between 4 and 11. Subjects with deviations in the optimal $N$ from this narrower range have essentially flat valleys between $N = 2$ and $N = 20$ (*Figure 4—figure supplement 1*), and thus the choice of $N$ is not strongly constrained.

The minimum fit errors are necessarily larger than the minimum fit errors for the across-subject averaged data, because of the higher variability of individual subject data (fewer trials per subject than total trials across subjects). Nevertheless, the normalized squared errors of the fits can be quite low, and the theory provides good fits to the psychophysics data for the individual subjects.

We also fit the individual subject data to the direct storage models, to be able to compare the predictions from the two models, *Figure 4—figure supplement 2*. We then compute the Bayesian Information Criterion score for both the direct storage model and the well-coded storage model, and report the $\Delta BIC$ score for hypothesis comparison, *Figure 4—figure supplement 2*. Positive (negative) $\Delta BIC$ scores indicate support for the well-coded (direct) storage model, and an absolute value of 10 or greater indicates very strong support. Note that the $\Delta BIC$ scores for the individual subjects are much smaller in magnitude than the aggregate scores for all pooled data in the main manuscript, because the data set for individual subjects is smaller and has less statistical strength. Nevertheless, there is very strong support ($|\Delta BIC| > 10$) for the well-coded model in 4 out of 10 subjects, close to strong support for direct storage in 2 out of 10 subjects ($|\Delta BIC| \approx 10$), positive support for direct storage in 2 subjects, and essentially insignificant support ($|\Delta BIC| \approx 2$) in 2 remaining subjects.

