## [Decision Letter]

Thank you for submitting your article "Fundamental bound on the persistence and capacity of short-term memory stored as graded persistent activity" for consideration by *eLife*. Your article has been reviewed by three peer reviewers, and the evaluation has been overseen by a Reviewing Editor and David Van Essen as the Senior Editor. The following individuals involved in review of your submission have agreed to reveal their identity: Tim Buschman (Reviewer #1); John D Murray (Reviewer #2); Brad Postle (Reviewer #3).

The reviewers have discussed the reviews with one another and the Reviewing Editor has drafted this decision to help you prepare a revised submission.

The manuscript presents an information-theoretic computational model of STM that suggests an intriguing new way that information may be coded in working memory. The theoretical framework developed here constitutes an important advance in linking neural circuit mechanisms to testable psychophysical behavior (here, working memory precision as a function of duration and load). The quantitative fit of the model to human behavior is compelling and bolsters the relevance of the theoretical advances.

The reviewers were all in agreement about the potential impact of this work presented. But all also agreed that further discussion of the proposed models and its implications should be added to more thoroughly place this work in the broader context of the field. Some specific suggestions are made below. Furthermore, there several detailed questions regarding aspects of the model that should also be addressed. I have edited and appended the revisions that are essential to include in a revision below.

Please address the following in a revision:

1) One reviewer noted that 'it takes too long to get to the point in the manuscript at which the reader knows, well, what the main point of the paper will be. It's not in the title, not in the Abstract, and, indeed, not clearly articulated until subsection “Information-theoretic bound on memory performance with well-coded storage” of the manuscript.' The first part of the manuscript is taken up with a lengthy exposition of why and how direct storage models are unsatisfactory. For a general-interest journal, one would want the central idea to be clearly articulated in one of the first paragraphs in the paper (not to mention in the Abstract), then the demonstration that direct storage models are insufficient to be dispatched within a few short paragraphs. Perhaps some of this could be accomplished in part by moving some of the text and analyses to figure legends? As things stand, the figures with their minimalist legends are inscrutable. One idea would be to display the panel from Figure 4 side-by-side with 3A and B, to permit a side-by-side comparison of the different approaches. Indeed, Figure 3 and Figure 4 could be merged, together with much of the text between them.

2) A second major absence from the Introduction, which will raise concerns by many familiar with the current literature, is near absence of any consideration of the growing number of suggestions that STM might be accomplished by mechanisms other than sustained activity. To name just a few, there's a recent TICS paper by Stokes that is explicitly devoted to this idea, there are several theoretical accounts by Tsodyks, Barack, and colleagues (nicely summarized in a recent Current Opinion review), and there's the nonlinear dynamical systems model of Lundqvist and colleagues, recently illustrated with data from Miller's group.

If some variant of these "activity-silent" accounts is correct, are the ideas presented in this manuscript irrelevant, or are there principles from the present theory that would apply? Additionally/alternatively, are there principles from the present theory that might apply to sustained activity supporting a behavior other than STM?

3) Some of the writing contains incomplete or misleading assertions. For example, the idea that there are constraints on the amount of time that information can be held in STM ignores the fact that a classically held hallmark of STM is precisely that it is not sensitive to the passage of time, per se. (Two examples are from Keppel and Underwood, and many demonstrations of prolonged retention of information in STM in anterograde amnesic patients.) Indeed, puzzlingly, one of the papers cited by the authors to substantiate their assertion is entitled "No temporal decay in verbal short-term memory."

4) The manuscript makes not contact with the growing literature of multivariate analyses of data from STM tasks, from nonhuman and human electrophysiology, and from human fMRI. Some of these studies show the ability to decode the contents of STM from delay-period activity with decoders trained on sample-evoked signal. Others suggest that the neural code may be dynamic, with minimal if any cross-temporal generalization (i.e., "off-diagonal" decoding). How does the proposed theory relate to this empirical literature? Without reference to these broader literatures, the present manuscript might be more suitable for a more specialized computational journal.

5) The authors argue that the currently accepted model of working memory predicts a linear increase in mean-squared error (MSE) over time and load (MSE ~ (load)*(time)). In contrast, they find a sub-linear increase in MSE with time (Figure 3). This sub-linearity is well fit by the well-coded model. However, some of this non-linearity could be due to other, less-capacity-limited, forms of memory at very short time delays. For example, iconic memory, thought to have an extremely high capacity, is likely still available at 100 ms (some might argue for longer). This could lead to a reduction in the MSE at the lowest time delays. Ideally the authors would control for this using masking stimuli. Alternatively, the authors could control for this by excluding the very short delays from the analysis (possibly increasing the maximum memory delay if needed for fits).

6) As with many working memory paradigms, it is not entirely clear how to define the working memory load in the current task. It seems subjects must remember multiple pieces of information per memorandum (e.g. both color and orientation) in all cases except for the single item. This would suggest memory load is actually 1, 4, 8, and 12. Does this non-linearity account for the poor fit of the linear "direct coding" model? It seems like it might not, given the poor fit in Figure 3 but it would still be worth testing the two models with different values for memory load. Similarly, recent work has suggested some degree of independence of working memory load across the two visual hemifields. Again, this would suggest only the balanced displays can be directly compared (e.g. 2, 4, and 6 items). Does the well-coded model still provide a better fit If the analysis is restricted to these three conditions?

7) The authors appropriately use BIC to perform model comparison. However, these model comparison criterion often penalize parameters to different degrees. Did the authors also find the well-coded model generalized to a withheld dataset better than the direct coding model?

8) Recent work has debated whether errors during working memory are due, in part, to guessing or not (e.g. Luck, Awh, Vogel, Bays, etc). In fact, Steve Luck argues for no increase in variance with load (or time?), instead only an increase in guess rate. If fitting a circular Gaussian to the distribution do the authors find an increase in variance or an increase in baseline (or both)? Related to this, it isn't clear to me how the pure 'sudden-death' framework matches with the diffusivity arguments made here. It seems that perhaps the well-coded model could explain the existence of complete failures to remember if the signal diffuses too much, but the model would still argue for some diffusion of memory over time. This doesn't seem consistent with the current model. I know the authors attempt to address this in the Discussion section of the current manuscript but I would encourage the authors to clarify their position.

9) This study uses the co-authors' human psychophysical data from Pertzov et al., 2016 Journal of Experimental Psychology. That study decomposed errors into three sources: (1) noisy representation; (2) mis-binding or non-target responses; and (3) random guessing. They reported that all three of these components increased with higher load and with longer delays. How does these prior findings relate to the present study? Are these different sources subsumed by the present model? Or are these important features that the present model (in the diffusive regime) does not account for? Does the present model produce only the first type of errors? The Authors mention that in another regime of the model, non-diffusive errors can produce pure guessing errors. Can the model speak to the mechanisms of mis-binding errors? Please include discussion of this point.

10) Regarding the implications for neural representations: The Authors discuss that one prediction of the model would be signatures of exponentially strong codes in neural representations. As I understand it, one way this could be implemented is that each of the N memory networks has a different spatial period for its periodic coding, as in the case of grid cells. The other feature of the present model is that for multi-item working memory, a memory network contains signals for all of the K items. It would be helpful if the Authors can clarify what the implications on neural representations are for this feature of distributed multi-item coding. Does this imply that single neurons would show mixed selectivity for multiple items? Please include discussion of this point.

---

## [Author Response]

Please address the following in a revision:1) One reviewer noted that 'it takes too long to get to the point in the manuscript at which the reader knows, well, what the main point of the paper will be. It's not in the title, not in the Abstract, and, indeed, not clearly articulated until subsection “Information-theoretic bound on memory performance with well-coded storage” of the manuscript.' The first part of the manuscript is taken up with a lengthy exposition of why and how direct storage models are unsatisfactory. For a general-interest journal, one would want the central idea to be clearly articulated in one of the first paragraphs in the paper (not to mention in the Abstract), then the demonstration that direct storage models are insufficient to be dispatched within a few short paragraphs. Perhaps some of this could be accomplished in part by moving some of the text and analyses to figure legends? As things stand, the figures with their minimalist legends are inscrutable. One idea would be to display the panel from Figure 4 side-by-side with 3A and B, to permit a side-by-side comparison of the different approaches. Indeed, Figure 3 and Figure 4 could be merged, together with much of the text between them.

We have now edited the Abstract and Introduction to convey what the manuscript is about much earlier in the text. Please see the new introductory paragraph: "In the present work, we make the following contributions: 1) Generate psychophysics predictions for information degradation as a function of delay period and number of stored items, if information is stored directly, without recoding, in persistent activity neural networks of a given size over given time interval; 2) Generate psychophysics predictions (though the use of joint source-channel coding theory) for a model that assumes information is restructured by encoding and decoding stages before and after storage in persistent activity neural networks; 3) Compare these models to new analog measurements \cite{Pertzov16} of human memory performance on an analog task as the demands on both maintenance duration and capacity are varied."

Please note that the early results of the manuscript are to establish the theoretical predictions for direct storage in persistent activity networks. To our knowledge, these predictions about degradation as a function of time and item number with direct storage have not been made explicit before, and so are one part of our results (if they had been made before it would have been easy to shorten this section and replace it with a citation). It is equally important to state the framework, formalism (including resource use parameters, etc.), and results for the direct storage model in the main results for comparison with the framework and parameters of the well-coced model, so that it is clear that we are making a fair comparison.

The figure captions are fairly long, and in merging plots as well as clarifying the captions as suggested, they have become slightly longer. Thus, moving more of the text of the results to the figure captions is not ideal. We have edited and shortened the direct storage Results section, but have not eviscerated it as we feel it is an integral part of our main result. As suggested, we have also merged Figure 3 and Figure 4, to make a direct comparison between the different models easier for the reader.

2) A second major absence from the Introduction, which will raise concerns by many familiar with the current literature, is near absence of any consideration of the growing number of suggestions that STM might be accomplished by mechanisms other than sustained activity. To name just a few, there's a recent TICS paper by Stokes that is explicitly devoted to this idea, there are several theoretical accounts by Tsodyks, Barack, and colleagues (nicely summarized in a recent Current Opinion review), and there's the nonlinear dynamical systems model of Lundqvist and colleagues, recently illustrated with data from Miller's group.If some variant of these "activity-silent" accounts is correct, are the ideas presented in this manuscript irrelevant, or are there principles from the present theory that would apply? Additionally/alternatively, are there principles from the present theory that might apply to sustained activity supporting a behavior other than STM?

We thank the reviewers very much for this comment. Indeed, we did not explicitly discuss activity-silent accounts of STM in our Introduction or Discussion (other than providing a reference to Mongillo and Tsodyks 2008, a model of how synaptic facilitation can aid in the robustness of short term memory). Given recent experimental and modeling results in this direction -- they are starting to form a compelling alternate STM mechanism to persistent activity mechanisms -- it is important to mention these accounts.

We have added a brief stand-alone passage in the Introduction, stating that our current work is complementary to efforts to explain STM in terms of synaptic facilitation/activity-silent mechanisms. In this passage, we cite the work of Mi, Katkov and Tsodyks, 2016: Barak and Tsodyks, 2014; Stokes, 2015 and Lundqvist et al., 2016.

With respect to the question about whether our model would apply to activity-silent mechanisms: In citing the model of Mongillo and Tsodyks, 2008 in our earlier manuscript, we had considered the possibility of synaptic facilitation as a source of a longer cellular time-constant to serve as the basis of STM, but we viewed that model as another persistent activity model, with activity supporting facilitation and facilitation supporting elevated activity. The facilitation process lent a slower intrinsic time-constant to the persistent activity feedback loop, thus providing a more robust/less fine-tuned way to generate persistent activity. Such a model would be subject to the same diffusion/drift problems as persistent activity models, qualitatively speaking (but quantitatively with lower noise or slower diffusion time-constant), and thus subject to similar degradation as considered in our present work.

The newer models cited in the paragraph may exhibit different dynamics, and be subject to different types of noise, in which case the general principle of restructuring of information to improve memory would still be true but the functional form of error versus number of items and *N* could be somewhat different. However, if the synaptic facilitation states in these models were subject to a Gaussian drift (e.g. if the facilitation states are analog-valued and some biophysical noise-process drives a random walk through the set of possible states even in the absence of neural activity), then they too could be could be treated as a bank of information channels with Gaussian noise and potentially our theory would extend to these, but with different parameters.

Since there are not yet good models of noise in the synaptic facilitation variable, for instance, and the effects of such noise on collective network memory states, we cannot directly yet compute a theoretical bound on memory performance for these mechanisms. However, that is definitely a future interest; with more theoretical work on modeling sources of noise in the activity-silent mechanisms, it will be possible to apply a similar theoretical framework to obtain bounds on memory performance with and without good encoding.

3) Some of the writing contains incomplete or misleading assertions. For example, the idea that there are constraints on the amount of time that information can be held in STM ignores the fact that a classically held hallmark of STM is precisely that it is not sensitive to the passage of time, per se. (Two examples are from Keppel and Underwood, and many demonstrations of prolonged retention of information in STM in anterograde amnesic patients.) Indeed, puzzlingly, one of the papers cited by the authors to substantiate their assertion is entitled "No temporal decay in verbal short-term memory."

Indeed, as the reviewers note, early studies have emphasized the temporal robustness of STM, and compared to “iconic” memory, STM is much less susceptible to forgetting. Consistent with this, our experimental results clearly demonstrate that single items are remembered with very little degradation over time, and the effects of increasing item number are stronger than the effects of increasing delay on memory performance.

However, there is performance degradation over time, especially for more items. We do not ourselves model pure temporal decay as a mechanism for memory loss, so it was not our intention to convey this in the Introduction. The source of confusion was our phrasing and references. We are now more careful in making a distinction between performance degradation over time versus the possible mechanisms for such degradation (which could include noise or interference or, less likely according to the literature, pure temporal decay mechanisms), please see our edits.

4) The manuscript makes not contact with the growing literature of multivariate analyses of data from STM tasks, from nonhuman and human electrophysiology, and from human fMRI. Some of these studies show the ability to decode the contents of STM from delay-period activity with decoders trained on sample-evoked signal. Others suggest that the neural code may be dynamic, with minimal if any cross-temporal generalization (i.e., "off-diagonal" decoding). How does the proposed theory relate to this empirical literature? Without reference to these broader literatures, the present manuscript might be more suitable for a more specialized computational journal.

Our formalism indicates that the representation within a memory channel must be in an optimised format, and that this format is not necessarily the same format that information was initially presented in. According to the information-theoretic view, the brain must perform a transformation from stimulus-space into an optimally coded form, and one might expect to observe this transition of the representation at encoding. The less optimal the original stimulus space, the more different the mnemonic code will likely be from the sample-evoked signal.

This insight by the reviewer constitutes a potential key prediction of the model, that in domains that are already combinatorially structured, neural representations should remain similar throughout the delay period, whereas in domains amenable to compression at encoding, neural codes during the delay will appear dynamic or at least different from the stimulus-evoked signal. We now include a discussion of this point in the manuscript (Discussion section, paragraph beginning "It remains to be seen whether neural representations for short-term visual memory are consistent …."), also citing papers in the literature that show variously show either stable, conserved coding during delay or varying, different states during delay.

5) The authors argue that the currently accepted model of working memory predicts a linear increase in mean-squared error (MSE) over time and load (MSE ~ (load)*(time)). In contrast, they find a sub-linear increase in MSE with time (Figure 3). This sub-linearity is well fit by the well-coded model. However, some of this non-linearity could be due to other, less-capacity-limited, forms of memory at very short time delays. For example, iconic memory, thought to have an extremely high capacity, is likely still available at 100 ms (some might argue for longer). This could lead to a reduction in the MSE at the lowest time delays. Ideally the authors would control for this using masking stimuli. Alternatively, the authors could control for this by excluding the very short delays from the analysis (possibly increasing the maximum memory delay if needed for fits).

Thank you for this comment. Please note that the real problem with the direct storage (linear) model is not so much that the function in time is linear, as that even the average slopes of the different-item number curves versus time are not fit by the slopes in the linear model: That is, if we fit the 1-item versus time data, then the predicted slope of the 6-item versus time slope prediction is far lower than the average slope of the actual data.

This can be seen in Figure 3. If we attempt to fit all the curves simultaneously as well as possible, again the slopes of the fits in time are far from the mean slopes of the curves, leaving aside the question of sub-linearity.

If we understand, the reviewer is suggesting the following scenario: Consider some process that has linear degradation of information in time (e.g. like direct storage of information into persistent activity networks). Add to this model the assumption that the 100 ms time-point is due to iconic memory. After excluding this 100 ms point, the uncoded model might provide a much better fit than it has so far, and it might also be more competitive with the coded model.

We now perform this analysis, and find that the uncoded model still fails to simultaneously fit the 1- and 6- item versus time data, and remains a substantially poorer fit than the coded model fit to the same data. The result does not change these qualitative comparisons.

6) As with many working memory paradigms, it is not entirely clear how to define the working memory load in the current task. It seems subjects must remember multiple pieces of information per memorandum (e.g. both color and orientation) in all cases except for the single item. This would suggest memory load is actually 1, 4, 8, and 12. Does this non-linearity account for the poor fit of the linear "direct coding" model? It seems like it might not, given the poor fit in Figure 3 but it would still be worth testing the two models with different values for memory load. Similarly, recent work has suggested some degree of independence of working memory load across the two visual hemifields. Again, this would suggest only the balanced displays can be directly compared (e.g. 2, 4, and 6 items). Does the well-coded model still provide a better fit If the analysis is restricted to these three conditions?

This is an excellent suggestion. We have now redefined the item numbers from (1, 2, 4, 6) to (1,4, 8, 12) and re-done the fits. We find that our qualitative conclusions remain unchanged.

7) The authors appropriately use BIC to perform model comparison. However, these model comparison criterion often penalize parameters to different degrees. Did the authors also find the well-coded model generalized to a withheld dataset better than the direct coding model?

Thank you for another good question. To address this, we redid the analysis by excluding one time-point across all item-number curves, then asked how well the curves obtained from fitting the other time-points predicted the error for the held-out data-point. We repeated this for another time-point. This is like a leave-one-out or jackknife cross-validation procedure. We find that the well-coded model predicts the withheld datapoints with smaller error than the uncoded/direct coding model.

8) Recent work has debated whether errors during working memory are due, in part, to guessing or not (e.g. Luck, Awh, Vogel, Bays, etc). In fact, Steve Luck argues for no increase in variance with load (or time?), instead only an increase in guess rate. If fitting a circular Gaussian to the distribution do the authors find an increase in variance or an increase in baseline (or both)? Related to this, it isn't clear to me how the pure 'sudden-death' framework matches with the diffusivity arguments made here. It seems that perhaps the well-coded model could explain the existence of complete failures to remember if the signal diffuses too much, but the model would still argue for some diffusion of memory over time. This doesn't seem consistent with the current model. I know the authors attempt to address this in the Discussion section of the current manuscript but I would encourage the authors to clarify their position.

Thank you for the opportunity to clarify. The direct storage model, which involves only diffusion, does not include a nonlinear "sudden-death" process. Instead, the error of recall will simply grow, continuously and monotonically, over time; it's still possible in this model that a noisy, discrete-in-time experiment will result in the appearance of a sudden-death event where there really is only continuous degradation in the underlying system (e.g. beyond some threshold of memory degradation, noise in the report or observation will make the memory appear to be "gone"). On the other hand, if information is stored in a well-coded way, according to some good error-correcting code, we would expect inherently sharp threshold behavior: such codes display a characteristic level of noise below which they can effectively suppress most error, and above which they are guaranteed to fail, and then their errors are large. Thus, the model would predict a relatively small accumulation of error over some interval, followed by a super-linear increase in squared error. We now clarify this point in the manuscript.

9) This study uses the co-authors' human psychophysical data from Pertzov et al., 2016 Journal of Experimental Psychology. That study decomposed errors into three sources: (1) noisy representation; (2) mis-binding or non-target responses; and (3) random guessing. They reported that all three of these components increased with higher load and with longer delays. How does these prior findings relate to the present study? Are these different sources subsumed by the present model? Or are these important features that the present model (in the diffusive regime) does not account for? Does the present model produce only the first type of errors? The Authors mention that in another regime of the model, non-diffusive errors can produce pure guessing errors. Can the model speak to the mechanisms of mis-binding errors? Please include discussion of this point.

Re. Misbinding: The current model does not address this source of error. We now clarify this fact in the text. Our model in its present form is presented a single feature dimension, in this case orientation, and thus it does not consider the binding problem/binding errors. Note that our model could in principle be extended to take into account the joint storage of pairs or more of features per item, by representing those features as part of a higher-dimensional continuous attractor network, as in the joint population code model considered by Matthey, Bays and Dayan (2015); this is certainly of future interest of us (but of course out of scope of the current work). We have now added a note about this point in the Discussion.

Re. sudden death: we have now clarified in Discussion how sudden death can be consistent with our framework: "In our framework, good encoding ensures that for noise below a threshold, the decoder can recover an improved estimate of the stored variable; however, strong codes exhibit sharp threshold behavior as the noise in the channel is varied smoothly. […]We note, however, that the fits to the data shown here were all in the below-threshold regime."

10) Regarding the implications for neural representations: The Authors discuss that one prediction of the model would be signatures of exponentially strong codes in neural representations. As I understand it, one way this could be implemented is that each of the N memory networks has a different spatial period for its periodic coding, as in the case of grid cells. The other feature of the present model is that for multi-item working memory, a memory network contains signals for all of the K items. It would be helpful if the Authors can clarify what the implications on neural representations are for this feature of distributed multi-item coding. Does this imply that single neurons would show mixed selectivity for multiple items? Please include discussion of this point.

Good question. It is difficult to imagine any scenario involving non-mixed selectivity for items in a strong-coding scheme, thus indeed mixed selectivity would be a prediction of such a such a scheme. We already had a longer discussion of the question of tuning curves for strong codes under the heading "Are neural representations consistent with exponentially strong codes?" in the Discussion. We have now added a comment about mixed selectivity there.